# Revealing the finite-frequency response
# of a bosonic quantum impurity

Sébastien Léger[1*], Théo Sépulcre[1*], Dorian Fraudet[1], Olivier Buisson[1],
Cécile Naud[1], Wiebke Hasch-Guichard[1], Serge Florens[1], Izak Snyman[2],
Denis M. Basko[3] and Nicolas Roch[1†]

**1** Univ. Grenoble Alpes, CNRS, Grenoble INP, Institut Néel, 38000 Grenoble, France
**2** Mandelstam Institute for Theoretical Physics, School of Physics, University of the
Witwatersrand, Johannesburg, South Africa
**3** Univ. Grenoble Alpes, CNRS, LPMMC, 38000 Grenoble, France

† nicolas.roch@neel.cnrs.fr

## Abstract

Quantum impurities are ubiquitous in condensed matter physics and constitute the most stripped-down realization of many-body problems. While measuring their finite-frequency response could give access to key characteristics such as excitations spectra or dynamical properties, this goal has remained elusive despite over two decades of studies in nanoelectronic quantum dots. Conflicting experimental constraints of very strong coupling and large measurement bandwidths must be met simultaneously. We get around this problem using cQED tools, and build a precisely characterized quantum simulator of the boundary sine-Gordon model, a non-trivial bosonic impurity problem. We succeeded to fully map out the finite frequency linear response of this system. Its reactive part evidences a strong renormalisation of the nonlinearity at the boundary in agreement with non-perturbative calculations. Its dissipative part reveals a striking many-body broadening caused by multi-photon conversion. The experimental results are matched quantitatively to a perturbative calculation based on a microscopically calibrated model. Furthermore, we push the device into a regime where perturbative calculations break down, which calls for more advanced theoretical tools to model many-body quantum circuits. We also critically examine the technological limitations of cQED platforms to reach universal scaling laws. This work opens exciting perspectives for the future such as quantifying quantum entanglement in the vicinity of a quantum critical point or accessing the dynamical properties of non-trivial many-body problems.



## Contents

* These authors contributed equally to the development of this work.

# 1 Introduction

The past years have seen many advances in the design of simulators for strongly interacting fermions and bosons, using cold atom lattices [1] and polariton fluids [2]. These platforms are indeed well suited to controllably explore the dynamics of bulk many-body problems [3, 4], especially in presence of collective phenomena such as the superfluid to Mott insulator transition [5, 6]. The many-body effects triggered by a discrete quantum system at the boundary of a quantum gas (also known as a quantum impurity) have also been investigated in a controlled fashion using various types of electronic quantum dots, leading to the observation of universal scaling laws [7, 8], and even more exotic quantum phase transitions [9, 10]. With some exceptions [11–13], most studies on quantum impurities focused on finite-voltage, zero-frequency

DC transport measurements, without the possibility to unveil the finite frequency dynamics of the impurity. In addition, quantum dot systems are strictly limited to fermionic environments, and proposals to realize analogous bosonic impurities in cold atoms [14, 15] have not succeeded so far. Nevertheless, bosonic impurity problems have triggered intense theoretical research over the years [16–19], initially motivated by fundamental aspects of quantum dissipation [20, 21]. Bosonic impurities can also be used to describe defects in strongly interacting bulk fermionic systems [22–24], using collective degrees of freedom [25]. The scarcity of controlled experiments in the many-body regime of bosonic impurity systems is therefore still a major issue.

It is thus clearly an important technological goal to engineer and characterize truly bosonic quantum impurities, and the simplest advocated path involves the coupling between an ultrasmall Josephson junction to a controlled electromagnetic environment. Such devices have been studied theoretically, either in the framework of the spin-boson model [26–32] in case of a capacitive coupling, or the boundary sine-Gordon (BSG) models [33, 34] in case of galvanic coupling, which will be the topic of our study. Indeed, the galvanic coupling limits the sensitivity of the system to fluctuating charges, which is also a nuisance for quantum simulators. The development of circuit quantum electrodynamics (cQED) [35] provides an ideal testbed for the design and the precise measurement of bosonic impurity models, thanks to the recent advances in accessing the finite-frequency response of microwave photons in the quantum limit. While their non-equilibrium dynamical properties open fascinating research directions [36, 37], equilibrium spectroscopic studies constitute an important milestone that is necessary to characterize those complex impurity systems, and have been made available only recently [38–41]. In fact, there are many predictions for the finite-frequency linear response of bosonic boundary models that still await some direct experimental measurement [25], let alone more subtle non-linear phenomena that were more recently considered in the cQED context [27, 29–31].

Our main focus here is to unambiguously characterize spectral signatures of non-perturbative effects in a prototypical bosonic quantum impurity problem described by the BSG Hamiltonian. For this purpose, we investigate non-linear effects that are controlled by a single Josephson junction at the edge of a high-impedance superconducting transmission line, combining state-of-the art cQED fabrication techniques and measurements, with fully microscopic many-body simulations. The use of high impedance meta-materials here is crucial to enhance the quantum fluctuations of the superconducting phase variable at the boundary, reaching regimes where linearized theories become invalid and striking many-body effects prevail. The experiment that we designed uses an array of 4250 Josephson junctions, so that our system is close to the thermodynamic limit [40]. As a result, a large number of electromagnetic modes can be resolved, which makes measurements based on phase shift spectroscopy [39] very accurate. In addition, we use here the flux tunability of a SQUID at the terminal junction (the bosonic impurity), which allows us to test for the first time some predictions for the renormalized scale of the BSG Hamiltonian. Our work complements several recent experimental and theoretical studies [41–45] that target non-linear effects in a bosonic impurity model at impedances equal to or larger than the superconducting resistance quantum $R_Q = h/(2e)^2 \simeq 6.5$ kΩ. Of particular relevance is [45] in which the relatively large Josephson to charging energy ratio $E_J/E_C$ of the terminal SQUID combined with an environmental impedance larger than $R_Q$ put the device in a regime where the losses are dominated by phase slips. We target here the exploration of quantum non-linear effects at somewhat lower dimensionless impedances $\alpha = Z/R_Q \simeq 0.3$ and at $E_J/E_C < 1$, where quartic (Kerr type) and higher order processes at the terminal junction are the dominant source of non-linear effects. In addition, the reactive response of the SQUID is studied in parallel. This allows to measure and explain both sides of the same problem for the first time with this type of system.

A bosonic impurity can couple single and multi-photon states of a multi-mode resonator. The resulting appearance of multi-photon resonances in linear response has been demonstrated experimentally [46]. An important physical outcome of our study is a demonstration that these processes can induce a significant many-body dissipation channel in its surrounding transmission line as was also observed in Refs. [40,45]. This effect can be dominant over other known loss mechanisms, either of extrinsic (e.g. loss inside the measurement line) or intrinsic origin (dielectric losses, or magnetic flux noise, see Appendix H). Our microscopic modelling is able to reproduce the measured many-body losses at high frequency in the regime where the Josephson energy of the terminal Junction is a small parameter that can be treated perturbatively. Ref. [45] studies a complementary regime where the $E_J/E_C$ ratio of the terminal junction is larger than 1. At $\alpha \gtrsim 1$, Ref. [45] matches measured many-body losses to theoretical predictions. Viewed together with our work, this demarcates the regime of $\alpha \lesssim 1$ and Josephson energy comparable to charging energy as the frontier for further theoretical work or quantum simulation. We also critically examine scaling predictions from universal models, and we provide a clear path for the future development of superconducting circuits in order to address universal transport signatures, a hallmark of strong correlations.

The manuscript is organized as follows. In Sec. 2, we present the boundary sine-Gordon (BSG) model and our design from a superconducting transmission line terminated by a flux-tunable SQUID. The microwave measurement setup is also introduced, together with the full microscopic model describing AC transport in the device. In Sec. 3, we present our main data and extract both reactive and dissipative responses from the finite frequency spectroscopy. In Sec. 4, we present various numerical calculations of these observables. A self-consistent theory, valid when the SQUID is threaded with magnetic fluxes close to zero, shows a drastic renormalization of the frequency at the bosonic boundary, allowing also to extract the unknown parameters of the device. In addition, a perturbative calculation, valid close to half flux quantum where the Josephson energy becomes a small parameter, is able to describe precisely the dissipative effects due to multi-photon conversion, which are shown to dominate the high frequency response of the junction. We conclude the paper of various perspectives that cQED techniques open for the simulation of strongly interacting bosonic phases of matter.

## 2 Tailoring the BSG simulator

### 2.1 Design principles

The BSG model describes the quantum dynamics of a resistively shunted Josephson junction, which can be referred to as the weak link, the impurity or the boundary. It has a Lagrangian:

$$L = L_{\text{env}} + \frac{\hbar^2}{4e^2} \frac{C_J}{2} (\partial_t \varphi_0)^2 + E_J \cos \varphi_0 \,. \tag{1}$$

The weak link has a critical current $2eE_J/\hbar$ and a shunting capacitance $C_J$ that accounts for charging effects when Cooper pairs tunnel through the junction. In an idealized description, $\varphi_0$ is viewed as the boundary value $\varphi_{x=0}$ of a continuous field $\varphi_x$, and the environment is described by a continuous relativistic quantum field theory:

$$L_{\text{ideal env}} = \frac{\hbar R_Q}{4\pi Z} \int_0^\infty dx \left[ 1/c(\partial_t \varphi_x)^2 - c(\partial_x \varphi_x)^2 \right], \tag{2}$$

where $c$ is the phase velocity in the environment. In the language of electronic circuits, this ideal environment is an infinite transmission line with an impedance $Z$ shunting the weak link that is constant at all frequencies, while $\hbar \partial_t \varphi_0 / 2e$ is the voltage across the weak link.

In principle, the capacitance $C_J$ provides an ultraviolet regularization: at high frequencies it shorts the circuit. However, the theoretical analysis of the BSG model often assumes that the environment has a finite plasma frequency $\omega_p$, that provides a lower ultraviolet cutoff than $E_C/\hbar = (2e)^2/\hbar C_J$. In this universal regime, it is well-established that the BSG model hosts a quantum phase transition. When $Z > R_Q$, environmentally induced zero-point motion delocalizes the phase $\varphi_0$, making it impossible for a dissipationless current to flow through the weak link. When $Z < R_Q$, on the other hand, a dissipationless current can flow. This corresponds to the "superconducting" regime where the phase $\varphi_0$ is localized in a minimum of the cosine Josephson potential. We will focus in this work on the "superconducting" phase where $Z < R_Q$, without making a priori assumptions about the shunting capacitance $C_J$, which will turn out to play an important role in the description of our experimental device.

Let us first gain a qualitative understanding of the system by expanding the Josephson cosine potential to the second order, which corresponds to replacing the weak link with a harmonic $LC$ oscillator of resonance angular frequency $\omega_J$ and characteristic impedance $Z_J$, still shunted by an environmental impedance $Z_{env}(\omega)$. When the weak link is fully decoupled from its environment, $\omega_J = 1/\sqrt{L_J C_J}$ and $Z_J = \sqrt{L_J/C_J}$, with $L_J = (\hbar/2e)^2/E_J$. Current biasing the junction means adding a source term $\hbar I(t)\varphi_0/2e$ to the Lagrangian, contributing to the voltage across the junction $\hbar \partial_t \varphi_0/2e$. According to the quantum fluctuation dissipation theorem then, at zero temperature, the phase fluctuations are given by:

$$\left\langle \varphi_0^2 \right\rangle = 2 \int_0^\infty d\omega \, \mathrm{Re} \, \frac{Z_0(\omega)}{\omega R_Q} \,, \tag{3}$$

where

$$Z_0(\omega) = \left[ \frac{1}{Z_J}\left( \frac{\omega}{i\omega_J} + \frac{i\omega_J}{\omega} \right) + \frac{1}{Z_{env}(\omega)} \right]^{-1} \tag{4}$$

is the impedance of the resistively shunted linearized weak link. In the ideal case where $Z_{env}(\omega) = Z$, this gives:

$$\left\langle \varphi_0^2 \right\rangle = \frac{Z}{R_Q} \int_0^\infty d\xi \frac{1}{\xi + \frac{Z^2}{Z_J^2}(\xi - 1)^2} \,. \tag{5}$$

When $Z_J \ll Z$, the weak link itself shorts the resistive shunt, so that the integrand in (5) develops a narrow resonance and $\left\langle \varphi_0^2 \right\rangle \simeq \pi Z_J/R_Q$ is very small, since we assumed $Z < R_Q$. When $Z = Z_J/2$, $\left\langle \varphi_0^2 \right\rangle = 4Z/R_Q$, while if $Z_J \gg Z$, the ohmic environment shorts the junction over a broad frequency range and $\left\langle \varphi_0^2 \right\rangle \simeq 4Z \ln(Z_J/Z)/R_Q$. These simple electrokinetic considerations dictate the design a non-trivial BSG simulator. Firstly, the observation of non-linear effects requires $\left\langle \varphi_0^2 \right\rangle \gtrsim 1$, hence $Z_J \gtrsim Z$. In addition, the Josephson term $E_J \cos(\varphi_0)$ contains all even powers of $\varphi_0$ and thus allows elementary processes in which a single photon disintegrates into any odd number of photons at $x = 0$. The largest signal of this disintegration is achieved when there is strong hybridization between the weak link and the environment. This requires that $Z_J$ is not too much larger than $Z$, and that $\omega_J$ is below the plasma frequency $\omega_p$ of the environment. The ideal situation is therefore to design a device where $Z_J \simeq Z$, with $Z$ on the order of (but smaller) than $R_Q$. A summary of the physical domains of BSG is already given in the right panel of Fig. 1, and will be discussed further in the text below.

## 2.2 The circuit

In our device, which is depicted in Fig. 1, the weak link is actually a SQUID consisting of two nearly identical small physical junctions on opposite sides of a ring. This results in a Josephson

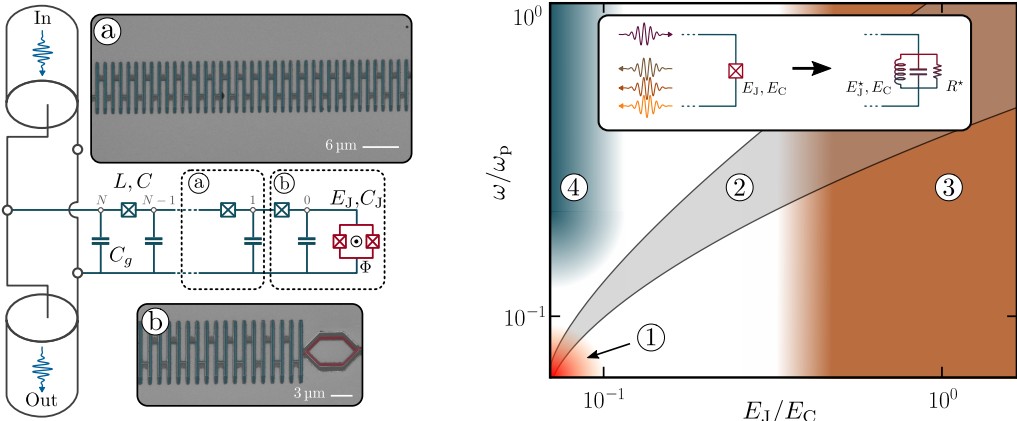

Figure 1: **Left.** Schematics of the measured circuit. The Josephson junction array, depicted in blue, is characterized by its lumped element inductance $L$, capacitance $C$ and ground capacitance $C_g$. The chain is terminated by a nonlinear SQUID, depicted in red and characterized by the flux-tunable Josephson energy $E_J(\Phi)$ and capacitance $C_J$. **a.** SEM picture of a small part of the full JJ chain, composed of 4250 sites in total. **b.** SEM picture of the galvanic coupling between the JJ chain and the nonlinear SQUID. The junction is grounded on its other side. **Right.** Parameter space of the device. The vertical axis represents the linear response probe frequency $\omega$ in units of the array's plasma frequency $\omega_p$. The horizontal axis represents the flux-tunable Josephson energy $E_J(\Phi)$ of the nonlinear SQUID in units of the SQUID's charging energy. The inset shows the two main BSG mechanisms: the reactive linear response is characterized by a renormalized Josephson energy $E_J^\star$, and further acquires a dissipative component $R^\star(\omega)$ due to photon disintegration at the boundary. Region **1** shows the low frequency universal limit of the BSG model, which is only a narrow domain of parameter space. The BSG mechanism produces many-body signatures across region **2**, which are strongest in a swathe around the SQUID's resonance frequency shown as grey shaded. In region **3** of moderate phase fluctuations, we constructed a microscopic mean field theory that accurately predicts the renormalization of $E_J^\star$. In the high frequency region **4**, we developed a perturbative microscopic method that accurately predicts the dissipative response of the device.

energy [47]:

$$E_J(\Phi) = E_J(0)\sqrt{\cos^2\left(\pi\frac{\Phi}{\Phi_Q}\right) + d^2\sin^2\left(\pi\frac{\Phi}{\Phi_Q}\right)}, \tag{6}$$

that can be tuned by varying the magnetic flux $\Phi$ through the SQUID ring. Here $\Phi_Q = h/2e$ is the flux quantum. In the above formula, $d$ characterizes the small accidental asymmetry of the SQUID, which is in the $10^{-2}$ to $10^{-1}$ range for the SQUIDS we fabricate. To estimate $E_J$, we constructed several isolated junctions using the same lithographic process as for our full device, and measured their room temperature resistance. Using the Ambegaokar-Baratoff law to extract the critical current, we find $E_J(0)/h \simeq 25$ GHz. The SQUID is further characterized by its internal capacitance $C_J$, predicted to be in the $10^1$ fF range from geometrical estimates.

To achieve an environmental impedance close to $R_Q$, we employ a long homogenous array of $N$ Josephson junctions. The junctions are large enough that they behave like linear inductors with inductance $L$ of order 1 nH. Such large junctions possess shunting capacitances $C$ in the $10^2$ fF range. The Josephson junctions separate $N+1$ superconducting islands, labeled 0 to $N$, each with a capacitive coupling $C_g \sim 10^{-1}$ fF to a back gate. At sufficiently low

frequencies, the inductance $L$ shorts the shunting capacitance $C$, and an infinite array of this type behaves like a transmission line with constant impedance $Z = \sqrt{L/C_g}$ in the targeted k$\Omega$ range. However, above a plasma frequency $\omega_p$ of the order of the resonance frequency $1/\sqrt{LC}$ of the capacitively shunted inductors, which is in the $10^1$ GHz range, electromagnetic excitations are unable to propagate down the chain. We employ an array with $N = 4250$ junctions. As explained in Appendix A, we were able to perform a sample characterization that yielded the following values for the array parameters: $L = 0.54$ nH, $C = 144$ fF, and $C_g = 0.15$ fF, so that $\omega_p = 2\pi \times 18$ GHz, and $Z/R_Q = 0.3$.

As seen in Fig. 1, the weak link connects to the rightmost node of the array. In order to perform spectroscopic measurements, the leftmost node is galvanically coupled to a $Z_{tl} = 50\,\Omega$ micro-strip feed-line, in a T-junction geometry. In Appendix B we provide an explicit expression for the environmental impedance $Z_{env}$ of the array coupled to the feed-line. Owing to the mismatch between the characteristic array impedance $\sqrt{L/C_g} = 1.9$ k$\Omega$ and that of the $50\,\Omega$ transmission lines, $Z_{env}$ has sharp peaks at frequencies corresponding to Fabry-Perot resonances in the array. More than 100 modes can be clearly observed below the plasma cutoff in our device, most of which lie in the ohmic regime, see the top panel of Fig. 2 showing a close-up on five of those modes. These modes are well-resolved, with a maximum free spectral range (level spacing) $\Delta f_{FSR} \simeq 0.4$ GHz that is larger than the line width of each mode. This provides a means to study the system response by spectroscopic analysis, as we detail now.

## 2.3 Measurement protocol

With the feed-line connected to the high impedance array as described in the previous section, we have the two port device with a hanging resonator geometry that is depicted in Fig. 1. We perform spectroscopic measurements on the hanging resonator by sending an AC microwave signal to the input of the feed-line and collecting the signal arriving at its output. The ratio between the complex amplitudes of these two signals defines the transmission of the circuit $S_{21}$. The microwave signal is calibrated to be sufficiently weak that no more than one incident photon on average populates the resonant modes of the circuit. Therefore, the circuit is close to equilibrium, so that standard linear response theory can be used for modelling its transport features. The array terminated by the weak link acts as a side-coupled resonator with a joint impedance to ground $Z_{a+w}$. Solving the classical scattering problem for electromagnetic waves in the feed-line in the presence of this resonator, yields that $S_{21} = 1/(1 + Z_{tl}/2Z_{a+w})$, which we can rewrite as

$$S_{21}(\omega) = \frac{2Z_N(\omega)}{Z_{tl}}. \tag{7}$$

Here $Z_N = (2/Z_{tl} + 1/Z_{a+w})^{-1}$ is the linear response impedance between superconducting island $N$ of the full device, where the array is connected to the feed-line, and the back gate (or ground). Note that (7) does not involve any semi-classical approximation. All quantum effects are included and $Z_N(\omega)$ is the linear response impedance obtained from the Kubo formula [27]. We discuss this further in Appendix B, where we provide an explicit formula relating $S_{21}$ to the system parameters and the self-energy associated with the weak link.

# 3 Finite frequency measurements

## 3.1 Spectroscopy of the device

When the probe frequency is close to one of the resonances $\omega_l$ of the circuit, the signal interferes destructively and a sharp drop in transmission is observed. Typical transmission curves as a function of the frequency are reported in the top panel of Fig. 2 for two different values

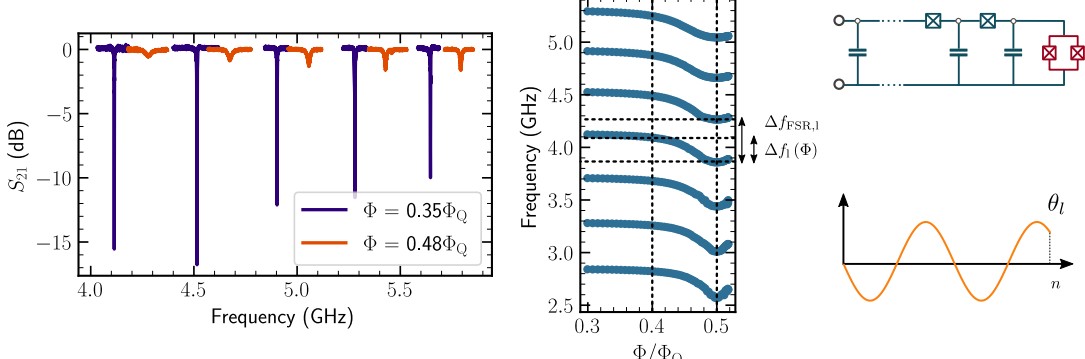

Figure 2: **Left.** Transmission $|S_{21}|$ as a function of the frequency at $\Phi/\Phi_Q = 0.35$ (violet) and $\Phi/\Phi_Q = 0.48$ (orange), showing an expected shift of the modes (due to the different boundary condition imposed by changes in the SQUID parameter), and a very dramatic reduction of the quality factor, that cannot be explained microscopically by a linearized description of the circuit. **Center.** Resonance frequencies $f_l$ as a function of the magnetic flux $\Phi/\Phi_Q$ for frequencies ranging from 2.5 to 5.4 GHz. The frequency shifts of the mode $l$ induced by the SQUID biased at flux $\Phi$ is labeled $\Delta f_l$. The free spectral range of mode $l$ is labeled $\Delta f_{\text{FSR},l}$. These two quantities are used to estimate the relative phase phase induced by the SQUID, defined in Eq. 9. **Right.** Phase mode profile for a mode $l$ as a function of the position index $l$. The phase shift $\theta_l$ induced at the SQUID boundary is defined in Eq. 8.

of the magnetic field. By tracking the resonance frequencies as function of flux through the SQUID, we obtain the lower panel of Fig. 2. The flux range here is limited to close to half a flux quantum due to the periodic dependence in magnetic field from Eq. (6). We note that the maximum amplitude of variation of each mode frequency is of the order of the free spectral range. This is due to the change of boundary condition, from nearly closed circuit (large $E_J$) at low flux to nearly open circuit (small $E_J$) at half flux. The only component of the device whose electronic properties has such a strong magnetic field dependence is the SQUID at the end of the array. We therefore conclude that our spectroscopic measurements are sensitive to the boundary term in our BSG simulator.

Besides this frequency shift of the modes, we further observe a striking flux dependence of the resonance widths. The frequency at which the largest broadening is observed decreases in a manner similar to the expected flux dependence of the SQUID resonance frequency, suggesting that the greatest broadening occurs for modes that are on resonance with the SQUID. (Not shown in Fig. 2, but see Fig. 3 below.) Furthermore, the average broadening of resonances is larger when the SQUID Josephson energy is lowest. These features are not reproduced when the SQUID is modelled as a linear circuit element as in (16), suggesting that the measured transmission contains significant information about inelastic photon processes due to the boundary SQUID. In the rest of the Article, we quantify these inelastic contributions to the measured signal and compare to theoretical predictions.

## 3.2 Shifting of the resonances

The dependence of resonance positions on the external magnetic flux reveals quantitative informations about the reactive effect of the weak link, which we can extract as follows [48]. The resonances we observe are predominantly single-photon in nature, and occur at photon wave vectors $k$ quantized such that $(N + 1/2)k_l(\Phi) + \theta_l(\Phi) = \pi(l + \frac{1}{2})$ where $\Phi$ is the external

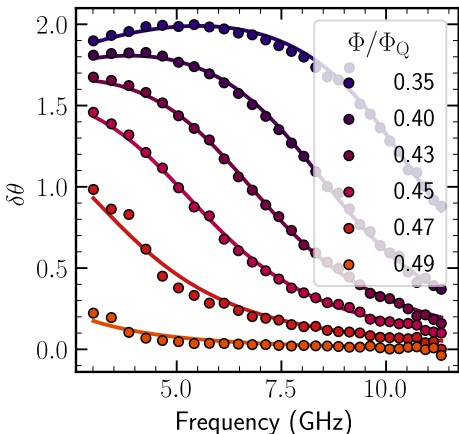

Figure 3: Phase shift $\delta\theta_l$ (relative to the limit $E_J \simeq 0$ taken at $\Phi/\Phi_Q = 1/2$) as a function of frequency $f_l$ of mode $l$, for six magnetic fluxes $\Phi/\Phi_Q$ taken between 0.35 and 0.49. Fitting the relative phase shift (dots) from a linearized model (lines) with effective inductance $L_J^\star$, we can deduce the renormalized Josephson energy $E_J^\star$ as a function of the magnetic flux.

flux, and $\theta_l(\Phi)$ is the phase shift associated with resonance $l$, which vanishes when the weak link is replaced by an infinite impedance. Provided that the asymmetry of the SQUID is small, we can take its effective inductance as infinite at $\Phi = \Phi_Q/2$. The relative phase shift

$$\delta\theta_l = \theta_l(\Phi) - \theta_l(\Phi_Q/2),\tag{8}$$

then measures the phase shift induced by the Josephson potential of the weak link. If we denote the $l^{\text{th}}$ resonance frequency as $f_l(\Phi)$ and notice that $\theta_{l+1} - \theta_l = \mathcal{O}(N^{-1})$ then it follows that

$$\frac{\Delta f_l(\Phi)}{\Delta f_{\text{FSR},l}} \equiv \frac{f_l(\Phi) - f_l(\Phi_Q/2)}{f_{l+1}(\Phi_Q/2) - f_l(\Phi_Q/2)} \simeq \frac{[k_l(\Phi) - k_l(\Phi_Q/2)]\partial_k f}{[k_{l+1}(\Phi_Q/2) - k_l(\Phi_Q/2)]\partial_k f} = \frac{\delta\theta_l(\Phi)}{\pi} + \mathcal{O}(N^{-1}),\tag{9}$$

where $\Delta f_l(\Phi)$ and $\Delta f_{\text{FSR},l}$ are respectively the frequency shift at $\Phi$ with respect to $\Phi_Q/2$ and the free spectral range for the mode $l$ (see Fig. 2). As was done in other recent recent experiments [39, 40] on dynamic quantum impurities in the context of superconducting circuits, we extracted the phase shift $\delta\theta_l$ as a function of external magnetic field and frequency.

To explore the relation between the phase shift and the properties of the weak link, we pose the following question: What would be the phase shift if the non-linear weak link was replaced by an effective linear inductor of inductance $L_J^\star = (\hbar/2e)^2/E_J^\star$, with $E_J^\star$ a renormalized Josephson energy? A formula for the phase shift in terms of $E_J^\star$, $C_J$ and the array parameters can be derived by finding the wave vectors $k$ where the impedance between node $N$ and ground, due to the array terminated in the weak link, vanishes. (See Appendix B for details.) In Fig. 3 we show the experimentally extracted relative phase shifts $\delta\theta_l$ as a function of mode frequency $f_l$, for various fluxes $\Phi$, together with the best fit to the effective linear theory (solid lines). We find excellent agreement between the experimentally extracted phase shifts and the theoretically predicted curve. We use the derived formula (B.10) in Eq. (8), and fit the extracted phase shifts $\delta\theta_l$ at different fluxes $\Phi$, the fitting parameters being $C_J$ and $L_J^\star(\Phi)$. (We use the array parameters quoted in Sec. 2.2.) From this, we extract $C_J = 14.5 \pm 0.2$ fF as well as the effective weak link Josephson energy $E_J^\star(\Phi) = (\hbar/2e)^2/L_J^\star(\Phi)$ as a function of flux. Its discussion is postponed to Sec. 4.

## 3.3 Broadening of resonances

The flux dependence of the broadening of spectroscopic resonances contains quantitative information about dissipation caused by photon disintegration in the weak link. We extract it as follows. Close to a resonance, the array together with the weak link has an impedance $Z_{\text{a+w},l} \simeq Z_{\text{dis},l} - iZ_{\text{reac},l} \times (\omega - \omega_l)/\omega_l$. Here $Z_{\text{dis},l}$ is real and caused by dissipation internal to the array or weak link. It does not have a significant frequency dependence on the scale of the resonance width. Similarly $-iZ_{\text{reac},l} \times (\omega - \omega_l)/\omega_l$ is the reactive (imaginary) part that vanishes at the resonance frequency $\omega_l$ and has been expanded in frequency around the resonance. From this follows that resonances in our setup then have the familiar 'hanging resonator' line shape,

$$S_{21}(\omega) = \frac{1 - 2iQ_{\text{i},l}\frac{\omega - \omega_l}{\omega_l}}{1 + \frac{Q_{\text{i},l}}{Q_{\text{e},l}} - 2iQ_{\text{i},l}\frac{\omega - \omega_l}{\omega_l}}, \tag{10}$$

where $Q_{\text{i},l} = Z_{\text{reac},l}/2Z_{\text{dis},l}$ and $Q_{\text{e},l} = Z_{\text{reac},l}/Z_{\text{tl}}$ are internal and external quality factors characterizing respectively dissipation in the weak link plus array, and in the feed-line. As a result, $1 - |S_{21}|^2$, as a function of frequency, has a Lorentzian line shape with halfwidth $(1/Q_{\text{i},l} + 1/Q_{\text{e},l})f_l$ in frequency.

 To study the internal dissipation due to the weak link, we therefore fit the hanging resonator line shape to individual resonances, and extract the internal broadening $\gamma_{\text{int},l} = f_l/Q_{\text{i},l}$ as a function of the resonance frequency and external magnetic field. In practice, connecting the feed-line to the array adds a small reactive part to the feed-line impedance. This causes a small peak asymmetry, which we include as another fitting parameter. Full details are provided in Appendix B. Besides the nonlinear processes taking place in the weak link, more mundane processes in the array can also contribute to $\gamma_{\text{int}}$. In recent years, several groups have been investigating the mechanisms that may be responsible for internal losses in superconducting resonators. For most materials [49], including resonators made out of Josephson junctions [50], the main mechanism that induces internal damping in the single-photon regime is the coupling with a bath formed by two-level-systems in dielectrics nearby the resonator. This effect, discussed further in Appendix G, does not depend on the external flux $\Phi$, and can thus be calibrated at $\Phi = 0$, where the non-linear contributions to the internal losses nearly vanishes, since the SQUID phase $\hat{\varphi}_0$ variable is well localized by the strong Josephson potential. It produces a constant-in-flux contribution $\gamma_{\text{diel}}$ that we subtract from the total internal broadening. In the results that we present below, we plot the resulting $\gamma_{\text{J}} = \gamma_{\text{int}} - \gamma_{\text{diel}}$ which represents the contribution of the broadening that is due to nonlinear effects due to the weak link only.

 We analyze the non-linear damping $\gamma_{\text{J}}$ of the modes due to the boundary junction as follows. In Fig. 4, we plot resonance frequencies between 2 GHz and 9 GHz as a function of flux $\Phi$. Each vertical column of data points in Fig. 4 represents a mode of the chain obtained from the frequency traces shown in Fig. 2. The color of each data point shows in log scale the broadening $\gamma_{\text{J}}$ due to the boundary junction, that we extracted by fitting the resonance line shape to (B.13), with the estimated dielectric losses in the chain subtracted. A dashed line indicates the effective weak link resonance frequency $\omega_{\text{J}}^{\star}(\Phi) = 1/\sqrt{C_{\text{J}}L_{\text{J}}^{\star}(\Phi)}$, with $L_{\text{J}}^{\star}(\Phi)$ as extracted from the phase shift data. We observe internal broadening varying from essentially zero (especially at fluxes close to zero where the system is nearly linear) to values exceeding 100 MHz, with excellent correlation between the effective weak link resonance frequency $\omega_{\text{J}}^{\star}$ and the maximum internal broadening. As the frequency cuts shown in the right panel of Fig. 4 reveal, $\gamma_{\text{J}}(\omega)$ decays exponentially for $\omega > \omega_{\text{J}}^{\star}$. The individual data sets with $\Phi/\Phi_0 > 0.45$ each show $\gamma_{\text{J}}$ decreasing by two decades as the probe frequency is scanned. This sharp frequency dependence within the physically accessible frequency window seems to be a unique property of the $E_{\text{J}} \ll E_{\text{C}}$ regime. In Ref. [45], where $E_{\text{J}}/E_{\text{C}} > 1$, $\gamma_{\text{J}}$ typically varies by less than one decade as the probe frequency is scanned.

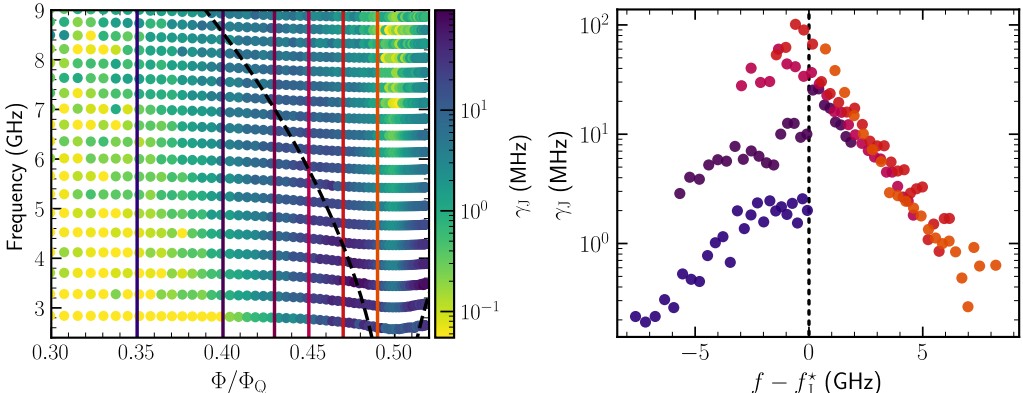

Figure 4: **Left.** Nonlinear damping $\gamma_J = \gamma_{\mathrm{int}} - \gamma_{\mathrm{diel}}$ of the modes due to the boundary junction as a function of magnetic flux $\Phi/\Phi_Q$ and mode frequency. The smallest dampings are in yellow while the largest are in blue. The dashed black line indicates the renormalized frequency $\omega_J^\star(\Phi)$ of the junction, while the full lines indicate the flux for which the nonlinear damping $\gamma_J$ are plotted in the right panel. **Right.** Nonlinear damping $\gamma_J$ as a function of $\omega - \omega_J^\star(\Phi)$ for various magnetic fluxes corresponding to the cuts (vertical lines) in the left panel. The color code is also the same as in Fig. 2. The damping rates are maximal around the renormalized SQUID frequency $\omega_J^\star$, and decay exponentially above this scale.

We have further estimated contributions to the internal broadening in the weak link itself due to other mechanisms not directly related to BSG physics. Obvious candidates are coupling with normal quasiparticles [51, 52] (*cf.* Appendix H) or inhomogeneous broadening from fluctuations in magnetic flux $\Phi$ through the SQUID. The latter, discussed in Appendix I, indeed depends on $\Phi$, but is sufficiently small to be discarded in our setup. It would furthermore produce a Gaussian line shape which is not what we observe. Normal quasi-particle tunnelling or dielectric losses in the SQUID both peak at frequencies close to the weak link resonance frequency. However, even under unrealistically favorable assumptions for these processes, they can contribute at most between $10^0$ and $10^1$ MHz to broadening (*cf.* Appendix H). We therefore conclude that the results in Fig. 4 are a clear manifestation of BSG physics in the weak link. The magnitude of the damping can be used to calculate the round-trip decay probability in the circuit for the single-photon excitations via:

$$p_{\mathrm{decay/RT}} = \frac{\gamma_J}{\Delta f_{\mathrm{FSR}}}, \tag{11}$$

and is equal to 0.25 for the maximal measured damping, a hallmark of ultra-strong coupling showing the large dissipation induced by the nonlinearity. In Ref. [45], similar round-trip decay probabilities are obtained in the transmon regime ($1 < E_J/E_C < 5$) and $\alpha \simeq 2$, while smaller decay probabilities were obtained in the transmon regime at $\alpha \simeq 0.7$. We now vindicate these qualitative effects by a microscopic modeling of the device.

# 4 Theoretical modelling of the observed many-body physics

Modelling theoretically the many-body effects in our experiment is challenging. An exact treatment is not feasible, due to the huge Hilbert space associated with the large number (a few hundreds) of modes that are involved in the ohmic range of the spectrum. However,

we can take advantage of the tunability of the SQUID junction to investigate in a controlled way the regimes of large and small Josephson energies $E_J(\Phi)$ of the boundary junction. We therefore discuss these two regimes separately.

## 4.1  Reactive effects at large $E_J$

At given $\alpha < 1$, fluctuations of the boundary phase $\varphi_0$ are controlled by the ratio $E_J/E_C$. Here we focus on the regime where $E_J/E_C$ is sufficiently large that phase fluctuations do not much exceed unity. The weak link has a charging energy $E_C$ of around $h \times 10$ GHz. At zero external flux, the weak link Josephson energy is more than twice as large, and the approximation in which the boundary Josephson energy is replaced by that of a linear inductor is adequate to capture the reactive aspects of the dynamics. Moving away from zero flux, a better approximation is to replace the bare value $E_J$ by a renormalized one $E_J^\star$, also called the self-consistent harmonic approximation (SCHA) [25, 53, 54]. This mean field theory is known to remain accurate at moderate phase fluctuations, when the phase explores more than the very bottom of the cosine Josephson potential, but does not tunnel out of the potential well. This regime corresponds to the region 3 of Fig. 1. To implement the SCHA for our circuit, we write the Josephson potential as

$$\hat{H}_{\mathrm{SQUID}} = \frac{E_J^\star(\Phi)}{2}\hat{\varphi}_0^2 - \left( E_J(\Phi)\cos(\hat{\varphi}_0) + \frac{E_J^\star(\Phi)}{2}\hat{\varphi}_0^2 \right). \tag{12}$$

The term in parenthesis is viewed as a perturbation that will be dropped, and $E_J^\star$ is chosen to make the resulting error as small as possible. This leads to the self-consistency conditions that the expectation value of the perturbation with respect to the ground state of the effective linear system should vanish. This self-consistency condition can be rewritten as

$$E_J^\star(\Phi) = E_J(\Phi)\exp\left(-\left\langle\hat{\varphi}_0^2\right\rangle/2\right), \tag{13}$$

where phase fluctuations $\left\langle\hat{\varphi}_0^2\right\rangle$ are computed using Eqs. (3-4) and the environmental impedance (B.7) derived in Appendix B with the effective junction impedance $Z_w = (\omega C_J/i + i/L_J^\star(\Phi)\omega)^{-1}$,[1] and $L_J^\star(\Phi) = (\hbar/2e)^2/E_J^\star(\Phi)$. For given bare Josephson energy $E_J(0)$ at zero flux and SQUID asymmetry $d$, the self-consistency condition (13) allows us to generate a curve $E_J^\star(\Phi)$, which is expected to be accurate at flux $\Phi$ not too close to $\Phi_Q/2$. We treat $E_J(0)$ and $d$ as free parameters and adjust this curve to the $E_J^\star(\Phi)$ data that we experimentally extracted with the aid of phase shift spectroscopy, see Sec. 3.2. We use data for $0.35 < \Phi < 0.47\Phi_Q$. This provides estimated values for the zero-flux bare $E_J(0)/h = 27.5$ GHz and SQUID asymmetry $d = 2\%$. The estimated asymmetry is reasonable for our fabrication process for small junctions given that we aimed for a perfectly symmetric SQUID. The value of $E_J(0)$ is in good agreement with the estimate of $E_J(0)/h = 25 \pm 1$ GHz obtained from the measurement of the room temperature resistance of isolated test junctions fabricated on the same wafer and at the same time as the full device. This confirms the accuracy of the SCHA at relatively large $E_J(\Phi)$.

The renormalization of $E_J$ is a textbook feature of the BSG model. Deep in the over-damped limit $Z_J \gg Z$, Eq. (5) yields $\left\langle\varphi_0^2\right\rangle \simeq 2\alpha \ln\left(E_C/(2\pi\alpha)^2 E_J^\star\right)$, with $\alpha = Z/R_Q$ the dimensionless resistance of the perfectly Ohmic environment. This is known as the scaling regime. Solving the self-consistency condition (13) then yields the well known scaling law:

$$E_J^\star = \mathrm{Min}\left( E_J, E_J\left[\frac{(2\pi\alpha)^2 E_J}{2E_C}\right]^{\frac{\alpha}{1-\alpha}} \right), \tag{14}$$

---

[1]Note the $e^{i\pi}$ factor with respect to the standard convention. This convention is used throughout the article.

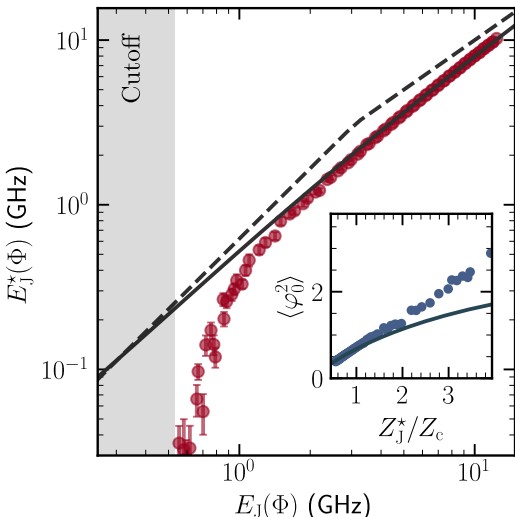

Figure 5: Extraction of the renormalized Josephson energy of the SQUID $E_J^\star(\Phi)$ as a function of the bare scale $E_J(\Phi)$, which is obtained from (6) by varying the flux $\Phi$. The dots correspond to the experimental data, the dashed line to the scaling law (14), and the full line to the complete SCHA solution with the microscopic circuit parameters. The inset displays the phase fluctuation of the SQUID $\langle\hat{\varphi}_0^2\rangle$ as a function of the renormalized Josephson impedance divided by the chain impedance $Z_J^\star/Z$. As expected, the quantum fluctuations increase with the SQUID impedance and reach several flux quanta for the largest impedance.

showing a strong downward renormalization of the Josephson energy $E_J^\star \ll E_J$ in the non-perturbative regime $0.1 < \alpha < 1$. Note that $E_J^\star$ cannot exceed the bare value $E_J$, which is why it has been bounded in Eq. 14. Note also that this scaling law predicts a superconducting to insulating Schmid transition at the critical value $\alpha = 1$, where the Josephson energy $E_J^\star$ renormalizes to zero due to a divergence of the phase fluctuations $\langle\hat{\varphi}_0^2\rangle$. At frequencies sufficiently below the plasma frequency, the weak link in our device sees an effective environmental impedance $Z_c \simeq \sqrt{L/C_g} = 1.9\,\mathrm{k\Omega}$ so that $\alpha = 0.3$. It is interesting to ask how the renormalization of $E_J^\star$ that we observe in our device compares to the renormalization predicted for an idealized system in the scaling regime.

In Fig. 5, we plot the observed renormalization $E_J^\star(\Phi)$ of the boundary Josephson energy in our BSG device, as a function of the bare scale $E_J(\Phi)$. The solid line shows the result that the fully microscopic SCHA calculation predicts for our device, and the dashed line shows the scaling law (14) for an idealized system with the same $E_C$ and $\alpha$ as in our device, but with infinite plasma frequency $\omega_p \to \infty$. We observe that $E_J^\star(\Phi)$ starts with a weak renormalization $E_J^\star/E_J \simeq 0.9$ at low flux (large bare $E_J$). Close to half flux quantum (small bare $E_J$), the measured renormalized scale becomes as small as $E_J^\star/E_J \simeq 0.2 \ll 1$. Except for this low flux regime, where it becomes invalid, the full SCHA provides an excellent description of the data. This underscores the fact that a detailed characterization of the environment is necessary in order to achieve agreement between theory and experiment in cQED simulators [39, 55]. By nature of the universal regime $E_J \ll E_C$, the analytical scaling law (14) should meet the full SCHA result at small $E_J$, which is what is seen indeed for $E_J(\Phi) < 1$ GHz. However, this is already the domain where the phase fluctuates very strongly, and both the SCHA and the scaling law are inapplicable.

Indeed, from the observed $E_J^\star(\Phi)$, we can estimate phase fluctuations using the self-consistency condition (13) as $\langle\hat{\varphi}_0^2\rangle = 2\ln(E_J/E_J^\star)$. We plot the estimated $\langle\hat{\varphi}_0^2\rangle$ as a function of the renormalized impedance $Z_J^\star = R_Q/2\pi\sqrt{4e^2/C_J E_J^\star(\Phi)}$ of the small junction in the inset of

Fig. 5. For flux $\Phi$ close to half flux quantum, the phase fluctuations increase up to the large value $\langle \hat{\varphi}_0^2 \rangle \simeq 4$, so that the phase $\varphi_0$ ventures far beyond the bottom of the cosine potential. The is is a clear indication of strong nonlinearities. The dissipative phenomena associated with the nonlinear dynamics of the weak link is explored in the next section.

## 4.2 Dissipative effects at small $E_J$

For $\alpha < 1$, the BSG model is known to flow to a Kondo-like strong coupling fixed point in the limit of zero temperature and for frequencies below a small emergent scale that characterizes the low-frequency inductive response of the weak link. The response of the system at these low frequencies are beyond the reach of a perturbative treatment [25]. Here we denote that scale $E_J^\star$ because in the universal regime $Z_J \gg Z$, it has the same scaling as in Eq. 14. Note however that the device we are modelling is not in the universal regime. Nonetheless we may expect $E_J^\star < E_J$. In order to tackle the strong non-linear regime of small Josephson energy, we develop perturbative theory that is controlled in the high frequency domain $\hbar\omega \gg E_J^\star$, using $E_J$ as a small parameter (compared to $E_C$ and $\hbar\omega_p$), which corresponds to the region 4 of Fig. 1. For simplicity, we outline here the zero-temperature calculation based on time-ordered Green's functions. Experiments on our device are performed at a temperature $T \simeq 30\,\text{mK} \simeq 0.6\,\text{GHz}$ (in units of $h/k_B$), that is of the same order as $E_J(\Phi_Q/2)$ and we therefore have to include finite temperature in our numerical calculations. The generalization to finite temperature is discussed in Appendix D.

If we set $E_J$ to zero, the impedance between the node zero of the array and ground is

$$Z_0^0(\omega) = \left[ \frac{1}{Z_{\text{env}}(\omega)} + \frac{|\omega|C_J}{i} \right]^{-1}. \tag{15}$$

An explicit expression (B.7) for the impedance $Z_{\text{env}}$ that shunts the weak link is provided in Appendix B. At zero temperature, this impedance is related to the time-ordered Green's function $G_{\varphi_0\varphi_0}(\omega) = -i \int_{-\infty}^{\infty} dt\, e^{i\omega t} \langle \mathcal{T} \hat{\varphi}_0(t)\hat{\varphi}_0(0) \rangle$ of the phase variable $\hat{\varphi}_0$ through

$$G_{\varphi_0\varphi_0}^0(\omega) = \frac{2\pi}{i|\omega|R_Q} Z_0^0(|\omega|), \tag{16}$$

(The superscript 0 of the Green's function indicates that it is calculated at $E_J = 0$). As discussed in Appendix B, the weak link self-energy, given by Dyson equation, $\Sigma(\omega) = \hbar[1/G_{\varphi_0\varphi_0}^0(\omega) - 1/G_{\varphi_0\varphi_0}(\omega)]$ fully captures the effect of the nonlinear Josephson potential on the linear response of the system at zero temperature. Indeed, $R_Q\hbar\omega/2\pi i\Sigma(\omega)$ enters the linear response functions we eventually wish to calculate as the impedance of a circuit element connecting node 0 of the array to ground.

At first sight, it seems that a straightforward expansion in powers of $E_J(\Phi)$ would allow us to calculate $\Sigma(\omega)$ for $\Phi$ in the vicinity of $\Phi_Q$ where $E_J(\Phi)$ is small. Indeed, dissipative effects show up at second order in $E_J$. However, because the system's response is nonperturbative at frequencies below an emergent scale $E_J^\star/h$, a regularization procedure is required in order to extract the response at frequencies in the measurement window between 2.5 and 11 GHz (well above $E_J^\star/h$) perturbatively. We first discuss the formal perturbative expansion of the self-energy and subsequently the regularization procedure.

To second order in $E_J$, we find the self-energy:

$$\Sigma(\omega) = E_J^v + i(E_J^v)^2 \int \frac{dt}{\hbar} \left[ \cos G_{\varphi_0\varphi_0}(t) - 1 + \frac{[G_{\varphi_0\varphi_0}(t)]^2}{2} \right]$$
$$+ (E_J^v)^2 \int \frac{dt}{\hbar} e^{i\omega t} \left[ \sin G_{\varphi_0\varphi_0}(t) - G_{\varphi_0\varphi_0}(t) \right]. \tag{17}$$

The vertex Josephson energy, is given by

$$E_{\text{J}}^{\text{v}} = E_{\text{J}} e^{-i G_{\varphi_0 \varphi_0}(t=0)/2} . \tag{18}$$

In Appendix C we present two independent derivations of this result.

In principle, a strict perturbative calculation would use the bare Green's function Eq. (16). Quite generally, formula (17) implies that the self-energy introduces linear response resonances associated with a single incoming photon disintegrating into multiple photons at the weak link. At zero temperature, these multi-photon resonances occur at frequencies that are sums of single-photon resonance frequencies. Owing to the approximately linear dispersion relation of the array $\omega = vk$ in the ohmic regime, single-photon resonance frequencies are almost equally spaced. As a result there is a large near-degeneracy in these multi-photon resonances. For instance, if we denote the lowest bare resonance frequency by $\omega_1$, then there are 16 multi-photon resonances, each involving an odd number of photons, at frequency $10\,\omega_1$, which corresponds to a single photon resonance in the middle of the experimentally accessible frequency window. This leads to a highly singular behaviour of the self-energy in the vicinity of these degenerate clusters of multi-photon resonances, when it is built on bare Green's functions. In our device this is not mitigated appreciably by the slight curvature of the photon dispersion or by geometric irregularity [46]. This singular behaviour is however spurious as it does not take into account the significant many-body level repulsion between multi-photon states coupled directly or indirectly by the highly non-linear terminal junction. We therefore self-consistently dressed all propagators with self-energy insertions to obtain what is also called a skeleton diagram expansion, or self-consistent Born approximation, which introduces many-body level repulsion and smoothens the self-energy. This is why we used the full interacting Green's function $G_{\varphi_0 \varphi_0}(\omega)$ in Eq. (17), which is determined self-consistently together with $\Sigma(\omega)$ from Dyson equation:

$$G_{\varphi_0 \varphi_0}(\omega) = \frac{1}{1/G_{\varphi_0 \varphi_0}^0(\omega) - \frac{1}{\hbar}\Sigma(\omega)} , \tag{19}$$

with $G_{\varphi_0 \varphi_0}^0(\omega)$ given by Eq. (16).

Let us now discuss the regularization procedure. When naively expanding in $E_{\text{J}}$ around zero, the Debye-Waller factor $E_{\text{J}}^{\text{v}}/E_{\text{J}} = \exp\left(-\langle \varphi_0^2 \rangle/2\right)$ is zero, due to a logarithmic divergence in $\langle \varphi_0^2 \rangle$. As a result, an unphysical answer $\Sigma = 0$ is obtained, so we do need to regularize the self-energy at low frequencies by introducing a counter-term $E_{\text{cutoff}}$,

$$\Sigma_{\text{reg}}(\omega) = \Sigma(\omega) - \Sigma(0) + E_{\text{cutoff}} , \tag{20}$$

where $E_{\text{cutoff}}$ must be larger than the true renormalized scale $E_{\text{J}}^{\star}$. The intuitive picture behind this regularization procedure is as follows. We imagine adding an extra linear inductor $L_{\text{cutoff}} = (\hbar/2e)^2/E_{\text{cutoff}}$ in parallel to the weak link to our model. It only adds a parabolic potential $E_{\text{cutoff}}\varphi_0^2/2$ that remains flat for $\varphi_0 = \mathcal{O}(1)$ to the Hamiltonian. At very low frequencies, this inductor shorts the weak link, thus providing an infrared regularization, but at frequencies in the measurement window, it hardly carries any current, and thus should not affect results. We have taken $E_{\text{cutoff}} = 0.05\,E_{\text{J}}(\Phi_Q/2) \approx 2\pi\hbar \times 0.02\,\text{GHz}$, and have checked that other choices of the same order of magnitude give the same results in the high frequency regime $\hbar\omega \gg E_{\text{J}}^{\star}$ where the calculation is controlled.

Using the expansion presented above, and considering again the bare Josephson energy $E_{\text{J}}$ and SQUID asymmetry $d$ as free parameters, we compare in Fig. 6 the measured internal linewidth (dots) to the theoretical predictions (lines), for three values of the flux $\Phi/\Phi_{\text{q}} = 0.48, 0.49, 0.5$. Note that two of those three curves are sufficient to fully determine $E_{\text{J}}$ and $d$, so that the theoretical curve at $\Phi/\Phi_{\text{q}} = 0.48$ contains no fitting parameter. The estimated values of the fitting

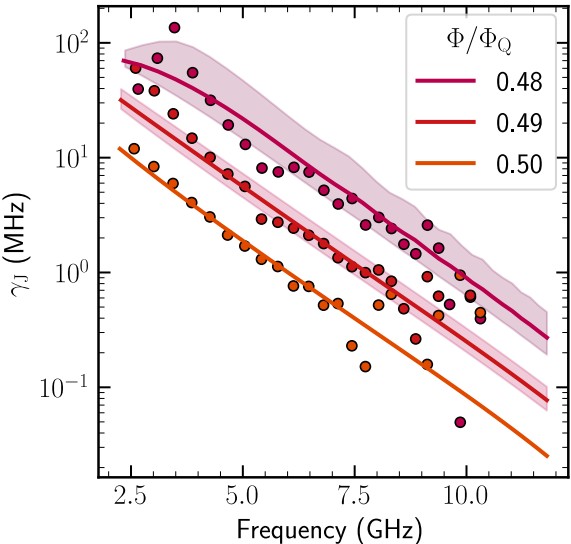

Figure 6: Many-body dissipation in the high frequency regime shown from the non-linear damping $\gamma_J(\omega_l)$ induced by the junction on mode $l$ of the chain, for three flux values $\Phi/\Phi_Q = 0.48, 0.49, 0.5$ (top to bottom). In this regime, the circuit obeys $E_J^\star \ll \hbar\omega$, allowing an expansion in powers of $E_J$. The dots correspond to the experimental data, the full lines are the theoretical prediction, and the shaded areas give the uncertainty on the fitted parameters. From these fits, both the bare Josephson energy $E_J$ and the SQUID asymmetry $d$ are extracted.

parameters are reported in Tab. 1 for the various measurements that have been performed (including the room temperature critical current and the fit of the renormalized scale $E_J^\star$), which give all very consistent results.

For smaller flux values, corresponding to the region 2 of Fig. 1, the junction frequency $\omega_J$ enters the measurement windows, and our theory surprisingly still describes qualitatively the maximum observed in the loss function $\gamma_J(\omega_k)$ for $\omega_l \simeq \omega_J^\star$. However, the magnitude of $\gamma_J$ is largely underestimated in the calculation, see Appendix F. This discrepancy is due to a breakdown of the expansion in powers of $E_J$, as we confirmed by computing all the order $E_J^3$ perturbative terms. At $\Phi < 0.4\Phi_Q$ we find that the $E_J^3$ contributions are of the same order as the $E_J^2$ contributions, while they remain negligible for the larger flux values of Fig. 6. At smaller fluxes, the superconducting phase is trapped near minima of the periodic Josephson potential, and non-perturbative $2\pi$-phase slip processes between minima provide the dominant contribution to the damping process, which are not taken into account in our perturbative treatment. Deviations from our model thus gives an estimate of these phase slip processes at $\alpha \lesssim 1$. These have also be investigated theoretically and experimentally at $\alpha \gtrsim 1$ [43–45].

Finally, we stress from figure 6 that the universal scaling law (See Appendix E) controlling the junction damping, $\gamma_J(\omega) \sim \omega^{2\alpha-2}$, is not obeyed in our measurement, rather an exponential

Table 1: Parameters estimated by three different methods.

| Method | $E_J/h$ (GHz) | d (%) |
|---|---|---|
| Room temperature resistance | 25.8(5) | - |
| Renormalization of the junction | 27.5 | 2 |
| Nonlinear loss of the junction | 25(3) | 2.4(4) |

decay is observed instead. This is expected because the scaling laws of the BSG model should be manifest on dynamical quantities only if $E_J^\star \ll \hbar\omega \ll E_C$, corresponding to the region 1 of Fig. 1. However, the charging energy $E_C$ of the junction is too small to fullfill both constraints together. The observed exponential decay can be explained qualitatively from the influence of the high energy cutoff on the photon conversion processes. When increasing the probe frequency, the number of available photonic states at higher frequency decreases exponentially (due to the reduction in combinatorics), drastically reducing the possibility of recombination of a single photon into various multi-photon states. One solution to observe the power law mentioned above would be to optimize the design of the boundary junction in order to increase $E_C$, but also to push the plasma frequency $\omega_p$ to higher values by increasing the transparency of the Josephson junctions in the chain, or by replacing them by a disordered superconductor. This would require important technological advances in the field of cQED.

## 5 Conclusion

In this work, we demonstrate a two-fold interplay between the boundary and the bulk from finite frequency measurements. On one hand, the bosonic environment induces a reactive response on the boundary degree of freedom, strongly renormalizing its resonance frequency. This effect is captured in the regime of large Josephson energy compared to its charging energy at the boundary junction, so that fluctuations of the superconducting phase variable remain moderate, and an effective linear model can apply (region 3 in Fig. 1). On the other hand, the boundary is also able to induce a dramatic dissipative response onto its environment, due to efficient frequency conversion into multi-photon states, which were shown to dominate over known sources of photonic losses. When the Josephson energy is small enough, it can be used as an expansion parameter, leading to a perturbative theory which accounts well for the measured high frequency response (region 4 in Fig. 1). Both approaches led to consistent estimates of the unknown parameters at the boundary junction. To compare our measurements using quantum many-body theory, we developed a fully microscopic model of the circuit. We found excellent agreement in regimes where the non-linear effects could be controlled. Interestingly, these two extreme regimes border a large domain of parameters where non-perturbative phenomena fully develop, and our circuit challenges all theoretical approaches we are aware of (region 3 in Fig. 1). We also evidenced that the use of universal scaling laws have to be taken with a grain of salt in superconducting circuits, due to the limited measurement bandwidth and relatively low ultraviolet cutoff set by the junction charging energy (a few GHz) and the plasma frequency (about 18 GHz). This scaling regime, where power laws in various response functions should develop, corresponds indeed to a parameter space that cannot be easily explored (region 1 in Fig. 1).

Future experimental developments of bosonic impurities in cQED could lead to the observation of more dramatic many-bodyl phenomena, such as quantum criticality [16], for instance the Schmid or spin-boson transitions that are predicted to occur at larger dissipation. Nevertheless, our work demonstrates that precursor effects of quantum phase transitions are worth investigating, because they exacerbate many-body behavior. Pushing further the cQED technology could also allow one to address bulk-dominated interactions close to the superconducting to Mott insulator transition of Josephson chains [56,57]. The direct observation of multi-photon conversion in cQED remains also a topic of interest, not only from the point of view of many-body physics [27,31], but also because they could be used as a potential quantum information resource.

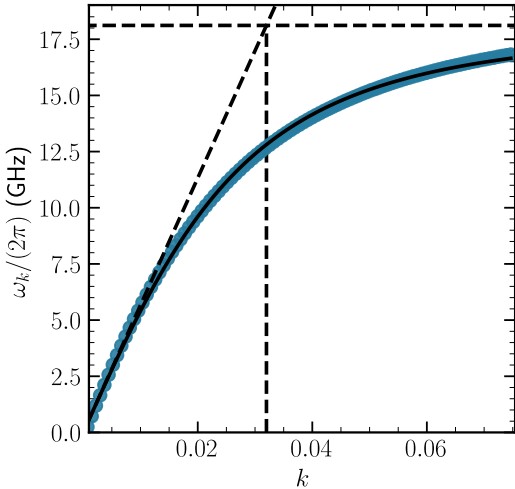

Figure 7: Dispersion relation of the array. Blue dots are the measured mode frequencies, and the full line results from the fit to Eq. (A.1). The horizontal dashed line is the plasma frequency $\omega_{\text{p}}$, the vertical one is $\sqrt{C_{\text{g}}/C}$ indicating when the Coulomb screening caused by $C$ starts. The linear line shows where the modes are TEM.

## A  Determining the array parameters

In order to find the chain parameters $C$, $C_{\text{g}}$ and $L$, we measure its dispersion relation using standard two tone spectroscopy, which allows us to accurately measure resonance positions from below 1GHz up to the plasma frequency at 19GHz The results as a function of wave number are shown in Fig. 7. For $k \ll 1$ the dispersion relation (A.1) is linear: $\omega_k = k/\sqrt{LC_{\text{g}}}$. For $k > \sqrt{C_{\text{g}}/C}$ on the other hand, the dispersion relation saturates to the plasma frequency $\omega_{\text{p}} = 1/\sqrt{L(C + C_{\text{g}}/4)}$. These asymptotic behaviors allow us to fit $C_{\text{g}}$ and $L$ once $C$ is known. We determine $C$ from knowledge of the area of the junctions composing the chain: $C = 45\text{fF} \times \text{area}[\mu\text{m}^{-2}]$ to set $C = 144\,\text{fF}$. From the fitting to the measured dispersion relation:

$$\omega_k = \frac{\sin\frac{k}{2}}{\sqrt{L\left(C\sin^2\frac{k}{2} + \frac{C_{\text{g}}}{4}\right)}} \implies k = 2\arctan\frac{\frac{\omega_k}{2}\sqrt{C_{\text{g}}L}}{\sqrt{1 - \frac{\omega_k^2}{\omega_{\text{p}}^2}}}, \tag{A.1}$$

we then estimate $L = 0.54\,\text{nH}$ and $C_{\text{g}} = 0.15\,\text{fF}$. Hence, we find the array impedance $Z_{\text{c}} = 1.9\,\text{k}\Omega$ and the plasma frequency $\omega_{\text{p}} = 18\,\text{GHz}$.

## B  Green's functions and impedance

The phase-phase correlation function and the linear response impedance play a crucial role in the analysis performed in this work. Here we elucidate their connection, and work out the various ingredients that are relevant for modelling our device. At zero temperature, it suffices to study time-ordered Green's functions, which is what we will discuss here for the sake of simplicity. At finite temperature, we employ (equilibrium) Keldysh Green's functions in our numerical computations. The necessary generalizations are discussed in Appendix D. Associated with the phase variables $\varphi_n$ on each island $n = 0, 1, \ldots, N$ of the array, we define the Green's function

$$G_{\varphi_m, \varphi_n}(t) = -i\langle \mathcal{T}\hat{\varphi}_m(t)\hat{\varphi}_n(0)\rangle, \tag{B.1}$$

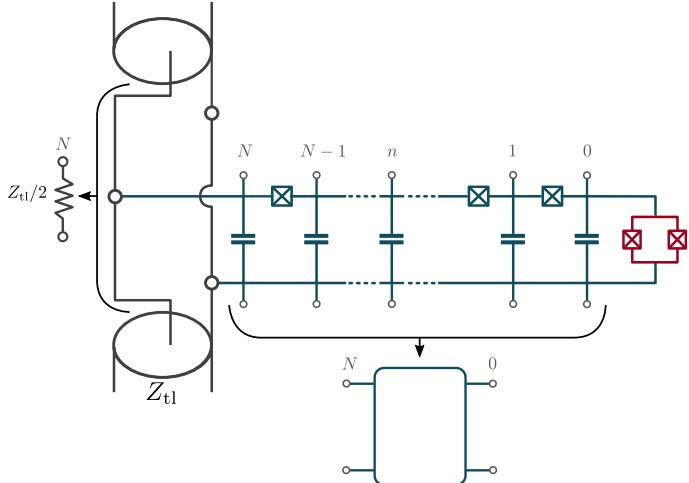

Figure 8: Our setup, viewed as an N+1 port device. To measure element $Z_{mn}$ of its impedance matrix one sends current into the top node and out of the the bottom node of port n, with all other ports open, and measures the voltage between the top and bottom node of port m.

the expectation value being with respect to the interacting ground state. The Fourier transform

$$G_{\varphi_m,\varphi_n}(\omega) = \int_{-\infty}^{\infty} dt\, e^{i\omega t} G_{\varphi_m,\varphi_n}(t) \tag{B.2}$$

has the property $G_{\varphi_m,\varphi_n}(-\omega) = G_{\varphi_n,\varphi_m}(\omega)$. At positive frequencies, $G_{\varphi_m,\varphi_n}(\omega) = G^{\mathrm{R}}_{\varphi_m,\varphi_n}(\omega)$, i.e. the time-ordered Green's functions, convenient for diagramatic expansions, are equivalent to retarded Green's functions that describe the system linear response. The identification between retarded Green's functions and impedance embodied in Eq.(16) in the main text extends to all superconducting islands in the array.

We view $G_{\varphi_m,\varphi_n}(\omega)$ as an $(N+1) \times (N+1)$ matrix $\mathbf{G}(\omega)$. The corresponding impedance matrix describes a $N+1$-port system obtained by associating a port with each superconducting island in the array, with one node of the port connected to the island, and the other to the back gate. See Figure 8. The operator corresponding to a current bias at port $n$ is $-\hbar I(t)\hat{\varphi}_n/2e$ while the voltage across port $m$ is $\hbar\partial_t\hat{\varphi}_m/(2e)$. Hence, at positive frequencies, $i\omega R_Q\mathbf{G}(\omega)/2\pi$ is the impedance matrix of the $N+1$-port system. The Dyson equation for $\mathbf{G}$ reads $\left\{[\mathbf{G}^0(\omega)]^{-1} - \mathbf{\Sigma}(\omega)/\hbar\right\}\mathbf{G}(\omega) = \mathbf{1}$. Here $\mathbf{G}^0(\omega)$ is the matrix Green's function when the weak link Josephson energy $E_J$ is set to zero. The self-energy $\mathbf{\Sigma}(\omega)$ incorporates the effect of the weak link. Since its energy $E_J(1-\cos\varphi_0)$ only involves the phase on island $n=0$,

$$\mathbf{\Sigma}(\omega)_{m,n} = \delta_{m,0}\delta_{n,0}\Sigma(\omega). \tag{B.3}$$

We thus identify

$$[i\omega R_Q\mathbf{G}^0(\omega)/2\pi]^{-1} + \frac{2\pi i\Sigma(\omega)}{\hbar\omega R_Q}, \tag{B.4}$$

at $\omega > 0$ as the $N+1$-port circuit admittance matrix in the presence of the weak link, while $[i\omega R_Q\mathbf{G}^0(\omega)/2\pi]^{-1}$ is the same, in the absence of the weak link. The fact that $\mathbf{\Sigma}(\omega)$ contributes additively to the admittance and has the form $\delta_{m,0}\delta_{n,0}\Sigma(\omega)$ then implies that as far as linear response is concerned, the effect of the weak link cosine potential is exactly equivalent to that of connecting a circuit element with impedance $R_q\hbar\omega/2\pi i\Sigma(\omega)$ across the nodes of port $n=0$.

Figure 9: Diagrams that show the circuits as well as the positioning of current sources and voltage probes used to define the impedances $Z_a$, $Z_{ab}$, $Z_{env}$ and $Z_{a+w}$. In each case $Z_X = V_X/I_X$.

Given the important role that impedance plays in our modelling, we now microscopically characterize the impedance of our device. We start by considering the array on its own, which we can view as an impedance network with a port at either end. One node of either port is connected to respectively the first or last superconducting island of the array and the other node of either port is connected to the back gate. The array $2 \times 2$ impedance matrix is

$$\begin{pmatrix} Z_a & Z_{ab} \\ Z_{ab} & Z_a \end{pmatrix} = \frac{2i \sin(k/2)}{\omega C_g \sin[(N+1)k]} \times \begin{pmatrix} \cos[(N+1/2)k] & \cos(k/2) \\ \cos(k/2) & \cos[(N+1/2)k] \end{pmatrix}, \quad \text{(B.5)}$$

with $k$ given by Eq. (A.1). See Fig. 9 for the definition of $Z_a$ and $Z_{ab}$. For completeness we mention that when $N \to \infty$, the resulting single-port element has impedance

$$Z_\infty(\omega) = \frac{1}{1-\omega^2 LC} \left( \sqrt{\frac{L}{C_g}} \sqrt{1 - \frac{\omega^2}{\omega_p^2}} + \frac{\omega L}{2i} \right), \quad \text{(B.6)}$$

which indeed reduces to $\sqrt{L/C_g}$ when $\omega \ll \omega_p$. The total environmental impedance that shunts the weak link is then that of the finite array connected to the feed-line at the far end (see Fig. 9),

$$Z_{env} = Z_a - \frac{Z_{ab}^2}{Z_{tl}/2 + Z_a}. \quad \text{(B.7)}$$

Another impedance of special significance is $Z_N(\omega)$, the impedance between array island $N$ and the back gate, which through Eq. (7) determines the measured transmission in the feed-line. Since at island $N$, the array is shunted by the transmission lines that carry the input and output signal,

$$Z_N = \frac{1}{2/Z_{tl} + 1/Z_{a+w}}, \quad \text{(B.8)}$$

where $Z_{a+w}$ is the impedance due to the array terminated in the weak link. In analogy to (B.7), it is given by

$$Z_{a+w} = Z_a - \frac{Z_{ab}^2}{Z_w + Z_a}, \quad \text{(B.9)}$$

which defines $Z_w$, the impedance of the weak link (see Fig. 9). If we model the weak link Josephson term as an effective linear inductor $L_J^\star(\Phi) = (\hbar/2e)^2/E_J^\star(\Phi)$, as within the self-consistent harmonic approximation (SCHA), then $Z_w \simeq \left[\omega C_J/i + i/\omega L_J^\star(\Phi)\right]^{-1}$. The relationship between the effective inductance $L_J^\star(\Phi)$ and the phase shift $\theta$ is obtained by setting $k = [\pi(l+\frac{1}{2}) - \theta(\Phi)]/(N+1/2)$ solving $Z_{a+w} = 0$ for $\theta$. This yields

$$\cot(\theta) = \frac{\omega\sqrt{C_g L}}{2\sqrt{1 - \frac{\omega^2}{\omega_p^2}}} \left( 1 - \frac{2L_J^\star(\Phi)}{L} \frac{1-\omega^2 LC}{1-\omega^2 L_J^\star(\Phi)C_J} \right). \quad \text{(B.10)}$$

Hence, we have an analytical expression for the relative phase shift defined in Eq. (8) where we approximate $L_J^\star$ to be infinite for $\Phi = \Phi_Q/2$.

If, on the other hand, we wish to include dissipation, we need to calculate the self-energy. The expansion is performed around the limit $L_J^\star \to \infty$. In this case the weak link linear response impedance can be related to the self-energy using Eqs. (B.4), (B.5):

$$Z_{\rm w} = \left[ \frac{\omega C_J}{i} + \frac{2\pi i \Sigma(\omega)}{R_Q \hbar \omega} \right]^{-1} . \tag{B.11}$$

Let us now investigate the line-shape of $S_{21} = 2Z_N/Z_{\rm tl}$ in more detail, including the small reactive contribution to the feed-line impedance, which we have ignored up to now. In the vicinity of a resonance, one can write

$$Z_N(\omega) \simeq \left[ \frac{2}{Z_{\rm tl} - iX} + \frac{1}{Z_{\rm dis} - iZ_{\rm reac}\frac{\omega-\omega_l}{\omega_l}} \right]^{-1} . \tag{B.12}$$

Here $(Z_{\rm tl} - iX)/2$ is the impedance due to the $50\,\Omega$ transmission lines carrying the input and output signals. Ideally this impedance would be purely real and equal to $Z_{\rm tl}/2 = 25\,\Omega$. In practice, the contact between the chain and the feed-lines contributes a small impedance $-iX/2$ in series, which is typically inductive ($X > 0$) and has a smooth frequency dependence on the scale of the free spectral range. Similarly, $Z_{\rm dis} - iZ_{\rm reac}(\omega - \omega_l)/\omega_l$ is the impedance of the array that terminates in the weak link, expanded to first order in both frequency around the resonance and real $Z_{\rm dis}$, the purely dissipative response at the resonance contained in $\Sigma(\omega)$. Here $-iZ_{\rm reac}(\omega - \omega_l)/\omega_l$ represents the reactive response in the vicinity of the resonance, i.e $Z_{\rm reac}$ is real. The parameters $X$, $Z_{\rm reac}$ and $Z_{\rm dis}$ can be taken as frequency-independent in the vicinity of a resonance. The resonances thus have the line shape

$$S_{21} = \left(1 - \frac{iX}{Z_{\rm tl}}\right) \frac{1 - 2iQ_{\rm i}\frac{\omega-\omega_l}{\omega_l}}{1 + \frac{Q_{\rm i}}{Q_{\rm e}}\left(1 - \frac{iX}{Z_{\rm tl}}\right) - 2iQ_{\rm i}\frac{\omega-\omega_l}{\omega_l}} , \tag{B.13}$$

where $Q_{\rm i} = Z_{\rm reac}/2Z_{\rm dis}$ and $Q_e = Z_{\rm reac}/Z_{\rm tl}$ are internal and external quality factors characterizing respectively dissipation in the weak link plus array, and in the external environment. This is the line-shape that we fit to the measured transmission resonances in order to extract the internal broadening $\gamma_{\rm int} = \omega_n/Q_{\rm i}/(2\pi)$ and the precise resonance frequencies $\omega_n/2\pi$.

## C Self-energy

Here we derive the self-energy expression (17) used in Sec. 4.2 to model the dissipative response of the BSG model. We perform the same calculation twice, using two equivalent approaches. In both cases, we perform Gaussian averaging of exponents whose arguments are linear in bosonic creation and annihilation operators. In the first derivation, we perform the required normal ordering by hand using Wick's theorem. In the second derivation, we represent the Wick contractions by Feynman diagrams. This is not as compact, but has the virtue of showing all multi-photon decay channels explicitly. The formal structure of our expansion is similar to that encountered for the bulk cosine nonlinearity in the Sine Gordon model so that the correctness of our result can be checked against results obtained in that context [58]. Subsequently, we discuss the self-consistent Born approximation.

### C.1 First derivation

Since the perturbation contains $\cos\varphi_0$, a diagramatic representation of the expansion to second order will already contain an infinite number of diagrams. Fortunately the amputation process can be automated as follows. In the path-integral language, and in the time

domain, the $G_{\varphi_0,\varphi_0}(t_2-t_1)$ Green's function, with external legs amputated, can be calculated from

$$G_{\text{amp}}(t) = i\left(\langle e^{iS_J}\rangle_0\right)^{-1}\left\langle\frac{\delta^2 e^{iS_J}}{\delta\varphi_0(t)\delta\varphi_0(0)}\right\rangle_0, \tag{C.1}$$

where $S_J = E_J\int_{-\infty}^{\infty}dt'\left[\cos\varphi_0(t')-1\right]/\hbar$ is the action associated with the weak link cosine perturbation, and $\langle\ldots\rangle_0$ denotes a Gaussian path integral over the field $\varphi_0(t)$, such that $-i\langle\varphi_0(t)\varphi_0(0)\rangle_0 = G^0_{\varphi_0,\varphi_0}(t)$. Without the functional derivatives, the right-hand side of Eq. C.1 would sum over all connected diagrams without external legs. The functional derivatives cut one bare $\varphi_0$ propagator of each such diagram, to produce two stubs, one at 0 and one at $t$, where external legs can be grafted. We expand (C.1) to second order in $E_J$ and go over from the path integral to the operator description where $\langle f[\varphi_0]\rangle_0 = \langle\mathcal{T}f[\hat{\varphi}_0]\rangle$, $f$ being any functional of the field $\varphi_0(t)$ and the right-hand side being the time-ordered expectation value of interaction-picture operators with respect to the zero-order ground state. We obtain

$$\begin{aligned}
G_{\text{amp}}(t) =& -\left\langle\frac{\delta^2 S_J}{\delta\varphi_0(t)\delta\varphi_0(0)}\right\rangle_0 - i\left\langle\frac{\delta^2 S_J}{\delta\varphi_0(t)\delta\varphi_0(0)}\left(S_J-\langle S_J\rangle_0\right)\right\rangle_0 \\
& -i\left\langle\left(\frac{\delta S_J}{\delta\varphi_0(t)}\right)\left(\frac{\delta S_J}{\delta\varphi_0(0)}\right)\right\rangle_0 \\
=& \frac{E_J}{\hbar}\langle\cos\hat{\varphi}_0\rangle\,\delta(t) + i\left(\frac{E_J}{\hbar}\right)^2\delta(t)\int_{-\infty}^{\infty}dt'\left\langle\mathcal{T}\cos\hat{\varphi}_0(0)\left[\cos\hat{\varphi}_0(t')-\langle\cos\hat{\varphi}_0\rangle\right]\right\rangle \\
& -i\left(\frac{E_J}{\hbar}\right)^2\langle\mathcal{T}\sin\hat{\varphi}_0(t)\sin\hat{\varphi}_0(0)\rangle.
\end{aligned} \tag{C.2}$$

Because the zero-order problem is harmonic, the field operator $\hat{\varphi}_0(t)$ is linear in boson creation and annihilation operators. One can expand the sin and cos functions of the field operators into exponentials. Under time-ordering, the product of exponentials of field operators equals the exponential of the sum of the operators. One is thus left with evaluating $\langle\mathcal{T}\exp X\rangle$ where $X$ is linear in boson creation and annihilation operators. It is well known that the result is $\langle\mathcal{T}\exp X\rangle = \exp\left(\langle\mathcal{T}X^2\rangle/2\right)$. Thus one straightforwardly obtains for instance

$$E_J^2\langle\mathcal{T}\sin\hat{\varphi}_0(t)\sin\hat{\varphi}_0(0)\rangle = i(E_J^{\text{v0}})^2\sin G^0_{\varphi_0\varphi_0}(t), \tag{C.3}$$

where

$$E_J^{\text{v0}} = E_J\exp\{-iG^0_{\varphi_0,\varphi_0}(0)/2\} \tag{C.4}$$

is the "tree-level" vertex energy. Calculating the remaining expectation values in (C.2) in a similar manner, we find

$$G_{\text{amp}}(t) = \frac{E_J^{\text{v0}}}{\hbar}\delta(t) + i\left(\frac{E_J^{\text{v0}}}{\hbar}\right)^2\delta(t)\int_{-\infty}^{\infty}dt'\left[\cos G^0_{\varphi_0\varphi_0}(t')-1\right] + \left(\frac{E_J^{\text{v0}}}{\hbar}\right)^2\sin G^0_{\varphi_0\varphi_0}(t). \tag{C.5}$$

At second order in $E_J$, the self-energy is related to the amputated Green's function through

$$\hbar G_{\text{amp}}(t) = \Sigma^{(1)}(t) + 1/\hbar\int dt'\,dt''\Sigma^{(1)}(t-t')G^0_{\varphi_0\varphi_0}(t'-t'')\Sigma^{(1)}(t'') + \Sigma^{(2)}(t), \tag{C.6}$$

where $\Sigma^{(n)}(t)$ is the $n$'th order in $E_J$ contribution to $\Sigma(t)$. We thus see that the linear in $G^0_{\varphi_0\varphi_0}(t)$ part of $\sin G^0_{\varphi_0\varphi_0}(t)$ corresponds to the second term on the right-hand side of (C.6) and that

$$\begin{aligned}
\Sigma^{(1)}(t) + \Sigma^{(2)}(t) =& E_J^{\text{v0}}\delta(t) + i\frac{\left(E_J^{\text{v0}}\right)^2}{\hbar}\delta(t)\int_{-\infty}^{\infty}dt'\left[\cos G^0_{\varphi_0\varphi_0}(t')-1\right] \\
& + \frac{\left(E_J^{\text{v0}}\right)^2}{\hbar}\left[\sin G^0_{\varphi_0\varphi_0}(t)-G^0_{\varphi_0\varphi_0}(t)\right].
\end{aligned} \tag{C.7}$$

## C.2 Second derivation

As an alternative to the above calculation, the self-energy can equivalently be represented diagramatically as follows. Expanding to second order, we construct all amputated diagrams with up to two vertices, that cannot be split in two by cutting an internal propagator:

$$-i(\Sigma^{(1)} + \Sigma^{(2)}) = \quad \text{(C.8)}$$

We have to organize this list. We focus on the effect of tadpoles. First, we single out a vertex. It can be dressed with any number of tadpoles, i.e. loops with a single propagator. We draw the rest of the diagram as a box, with any even number of lines between it and the singled out vertex. The sum over tadpoles reads

$$= \left(1 - \frac{iG^\circ(0)}{s_1} + \frac{(iG^\circ(0))^2}{s_2} - \dots\right) = \quad, \quad \text{(C.9)}$$

where the whole sum as been absorbed into a new vertex, depicted as a grey disk. The factorization above worked because symmetry factors are multiplicative: if $s$ is the symmetry factor of a diagram, the same diagram with $n$ more tadpoles on some vertex will have symmetry $s \times 2^n n!$. Thus, the dressed vertex equals $E_J^{V0}$ of Eq. C.4. Using this vertex dressing, we reduced the list of diagrams to,

$$-i(\Sigma^{(1)} + \Sigma^{(2)}) = \quad \text{(C.10)}$$

The symmetry factors in the second line and third lines work out such that the sums become respectively the cosine and sine of the propagator, with the leading term removed. Thus the second and third lines above exactly correspond to the second and third lines in Eq. C.7.

Because the Hamiltonian is even in the phases, and in particular in $\varphi_0$, photon number parity is conserved, that is, one photon can decay into an odd number of photons only. This selection rule is fully respected by our calculation, and the diagrammatic representation of the self-energy is especially convenient to see this fact. Namely, non-linear vertices can have only an even number of legs, and in the $\sin(G) - G$ self-energy term which is responsible for the decay, each vertex (0 or $t$) has one and only one external leg, so the two vertices must be connected by an odd number of photonic lines.

## C.3 Self-consistent Born Approximation

We can sum over a larger subset of diagrams by dressing the zero-order propagators appearing in the above result by all possible self-energy insertions. This takes into account that when a

photon disintegrates at the weak link, it does not disintegrate into bare photon modes of the harmonic system, but into modes that are themselves hybridized with the weak link. If we ignore this dressing of propagators in the self-energy, the approximate interacting Green's function contains resonances when an incoming photon has a frequency equal to the sum of any (odd) $n$ single-photon resonances of the harmonic zero-order problem. Given the nearly linear dispersion relation at low frequencies, this incorrectly predicts dense clusters of nearly degenerate $n$-photon resonances.

The dressing of propagators in the self-energy incorporates the level-repulsion between these resonances, which spreads them out over the free spectral range, thus giving a smooth background, rather than pronounced many-body peaks. The dressing of propagators in the self-energy leads to the replacement $G^0_{\varphi_0\varphi_0}(t) \to G_{\varphi_0\varphi_0}(t)$ in (C.7). However, if this is done blindly, there will be double-counting of some diagrams. For instance, because $G_{\varphi_0\varphi_0}(0) = G^0_{\varphi_0\varphi_0}(0) + E^{\text{v0}}_{\text{J}} \int dt' \, G^0_{\varphi_0\varphi_0}(t')^2/\hbar + \ldots$, when we dress the term $E^{\text{v0}}_{\text{J}}\delta(t) = E_{\text{J}}\exp[-iG^0_{\varphi_0\varphi_0}(0)/2]\delta(t)$ and expand the exponential around $-iG^0_{\varphi_0\varphi_0}(0)/2$, we encounter a term $-i(E^{\text{v0}}_{\text{J}})^2\delta(t)\int dt' \, G^0_{\varphi_0\varphi_0}(t')^2/\hbar$ which equals the quadratic part of $i\left(E^{\text{v0}}_{\text{J}}\right)^2\delta(t)\int_{-\infty}^{\infty}dt'\left[\cos G^0_{\varphi_0\varphi_0}(t')-1\right]/\hbar$. This happens because the full propagators used in the self-energy, built on a self-energy expansion to first order, already incorporates in an exact manner any terms in the perturbation that are quadratic in $\varphi_0$. To cure the double-counting, we must therefore remove the second-order in $G$ part of the cosine term in the dressed self-energy. Thus we arrive at the dressed self-energy of Eq. (17) in the main text, in which the dressed Green's function must be found self-consistently from the Dyson equation (19) in the main text.

# D  Keldysh technique

Finite temperature results are often obtained from diagramatic calculations that employ imaginary time Green's functions. To obtain the retarded Green's function, one has to perform an analytic continuation from imaginary to real time. This step is hard to perform numerically at the desired spectral resolution for our system with its many sharp resonances. We therefore rather use the Keldysh formalism in equilibrium, which does not involve such analytic continuation, to compute retarded Green's functions at finite temperature.

Instead of the time-ordered Green's function we employed previously, we have to use the contour-ordered Green's function, $\mathbb{G}_{\sigma\sigma'}(t,t') = -i\left\langle \mathcal{T}_c\hat{\varphi}_0(t)\hat{\varphi}_0(t')\right\rangle$. Here $\mathcal{T}_c$ orders operators along a time contour with a forward branch $\sigma = +$ from $-\infty$ to $\infty$ and a backward branch $\sigma = -$, from $\infty$ to $-\infty$. If $(\sigma,\sigma') = (+,+)$ the largest time of $t$ and $t'$ is to the left. If $(\sigma,\sigma') = (-,-)$, the largest time of $t$ and $t'$ is to the right. If $(\sigma,\sigma') = (-,+)$, $t$ is to the left, while if $(\sigma,\sigma') = (+,-)$, $t'$ is to the left. The expectation value refers to a thermal average. Thanks to the close similarities between contour ordering and time-ordering, the self-energy for $\mathbb{G}$ can be calculated using the same machinery as in Appendix C. In the path integral language there are independent fields $\varphi_{0\pm}(t)$ associated with the forward and backward branches of the time contour. The weak link action is $S_{\text{J}} = E_{\text{J}} \int_{-\infty}^{\infty} dt \left[\cos\varphi_{0+}(t) - \cos\varphi_{0-}(t)\right]/\hbar$. The four components of the self-energy are extracted by applying functional derivatives with respect to forward and backward fields

$$i\left\langle e^{iS_{\text{J}}}\right\rangle_0^{-1}\left\langle \frac{\delta^2 e^{iS_{\text{J}}}}{\delta\varphi_{0\sigma}(t)\delta\varphi_{0\sigma'}(t')}\right\rangle_0, \tag{D.1}$$

in analogy to the calculation in Appendix C. Summing the same class of diagrams as in Ap-

pendix C, one obtains

$$
\begin{aligned}
\Sigma_{\sigma,\sigma'}(t,t') =& \sigma \delta_{\sigma\sigma'} E_J^v \delta(t-t') \\
&+ i\sigma\delta_{\sigma\sigma'} \frac{(E_J^v)^2}{\hbar} \delta(t-t') \times \int_{-\infty}^{\infty} dt'' \sum_{\sigma''} \sigma'' \left[ \cos \mathbb{G}_{\sigma\sigma''}(t,t'') + \frac{\mathbb{G}_{\sigma\sigma''}(t,t'')^2}{2} \right] \\
&+ \sigma\sigma' \frac{(E_J^v)^2}{\hbar} \left[ \sin \mathbb{G}_{\sigma\sigma'}(t,t') - \mathbb{G}_{\sigma\sigma'}(t,t') \right].
\end{aligned} \tag{D.2}
$$

The vertex energy is

$$
E_J^v = E_J \exp[-\langle \hat{\varphi}_0^2 \rangle /2] = E_J \exp[-i\mathbb{G}_{\sigma\sigma'}(t,t)/2], \tag{D.3}
$$

(Any component of $\mathbb{G}_{\sigma\sigma'}$ can be used in the vertex energy, since they are all equal at coinciding times). In equilibrium, $\Sigma_{\sigma\sigma'}(t,t')$ and $\mathbb{G}_{\sigma\sigma'}(t,t')$ only depend on the time difference $t-t'$, and can be Fourier-transformed from time-difference to frequency. The self-energy $\Sigma$ and the Green's function $\mathbb{G}$, viewed as $2 \times 2$ matrices with entries arranged according to $\begin{pmatrix} ++ & +- \\ -+ & -- \end{pmatrix}$, must be found self-consistently from Eq. (D.2) together with the Dyson equation in matrix form:

$$
\left\{ \left[ \mathbb{G}^0(\omega) \right]^{-1} - \frac{1}{\hbar} \Sigma(\omega) \right\} \mathbb{G}(\omega) = \mathbb{1}. \tag{D.4}
$$

In equilibrium, the contour-ordered Green's function is related to the retarded Green's function through

$$
\begin{pmatrix} G^K_{\varphi_0,\varphi_0} & G^R_{\varphi_0,\varphi_0} \\ G^A_{\varphi_0,\varphi_0} & 0 \end{pmatrix} = \frac{1}{2} \begin{pmatrix} 1 & 1 \\ 1 & -1 \end{pmatrix} \mathbb{G} \begin{pmatrix} 1 & 1 \\ 1 & -1 \end{pmatrix}, \tag{D.5}
$$

with $G^A_{\varphi_0,\varphi_0}(\omega) = G^R_{\varphi_0,\varphi_0}(\omega)^*$ and $G^K_{\varphi_0,\varphi_0}(\omega) = 2i \operatorname{Im} G^R_{\varphi_0,\varphi_0}(\omega) \coth \frac{\hbar\omega}{2k_B T}$. The zero-order retarded Green's function reads [cf. Eq. (16)]

$$
G^{R0}_{\varphi_0,\varphi_0}(\omega) = \frac{2\pi}{iR_Q\omega} \left[ \frac{1}{Z_{\text{env}}(\omega)} + \frac{\omega C_J}{i} \right], \tag{D.6}
$$

so that

$$
\begin{aligned}
\left[ \mathbb{G}^0(\omega) \right]^{-1} = \frac{i\omega R_Q}{2\pi} \Bigg[ & \coth \frac{\hbar\omega}{2k_B T} \operatorname{Re} \frac{1}{Z_{\text{env}}(\omega)} \begin{pmatrix} 1 & -1 \\ -1 & 1 \end{pmatrix} \\
&+ \begin{pmatrix} \frac{\omega C_J}{i} + i\operatorname{Im}\frac{1}{Z_{\text{env}}(\omega)} & \operatorname{Re}\frac{1}{Z_{\text{env}}(\omega)} \\ -\operatorname{Re}\frac{1}{Z_{\text{env}}(\omega)} & -\frac{\omega C_J}{i} - i\operatorname{Im}\frac{1}{Z_{\text{env}}(\omega)} \end{pmatrix} \Bigg].
\end{aligned} \tag{D.7}
$$

After the self-energy $\Sigma$ of the contour-ordered Green's function has been calculated, the retarded self-energy (that enters the admittance matrix) can be extracted through

$$
\Sigma(\omega) = \frac{1}{2} \left[ \Sigma_{++}(\omega) + \Sigma_{+-}(\omega) - \Sigma_{-+}(\omega) - \Sigma_{--}(\omega) \right]. \tag{D.8}
$$

This self-energy is then substituted into Eq. (B.11) for the weak link impedance when the measured transmission signal is calculated.

# E  Scaling laws

In the main text, we remarked that the experimentally observed internal broadening of the chain modes decays exponentially as a function of frequency above $\omega_J^\star$. This is also what our

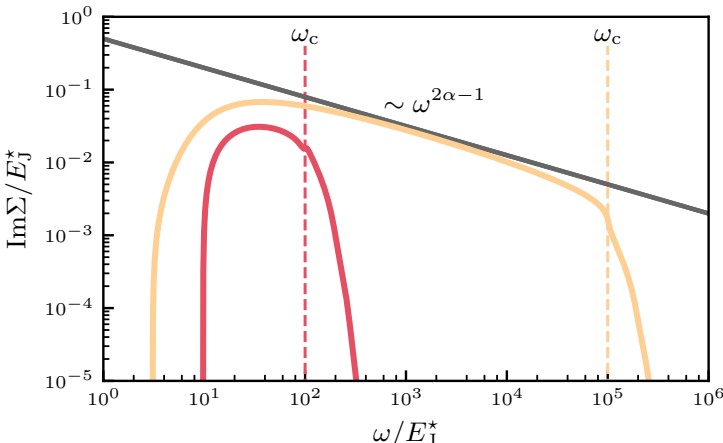

Figure 10: Dissipative part of the self-energy simulated with the same parameter as the device, pushing the ultra-violet cutoff $\omega_c$ to respectively $10^2$ and $10^5$ times the renormalized scale $E_J^\star$. Only in the latter unrealistic case can a clear power-law $\omega^{2\alpha-1}$ scaling be witnessed, while in the former case, an exponential suppresion is rather observed, in agreement with the experimental measurement.

microscopic theory predicts. However, in the theoretical literature, the BSG model is more usually associated with power-law dissipation. Here we review how the power law comes about, and explain why experimental realizations in cQED will have a hard time to exhibit such behavior.

The starting point is to assume that fluctuations of $\varphi_0$ are determined by the environmental impedance over a broad frequency range. In other words $Z_J \gg Z_{env}$ or $\hbar\omega_p \ll (2e)^2/C_J$. Approximating the environment as an Ohmic impedance $Z_{env}(\omega) = \alpha R_Q$, yields a zero order (time-ordered, zero-temperature) Green's function

$$G^0_{\varphi_0\varphi_0}(\omega) = \frac{1}{\frac{i|\omega|}{2\pi\alpha} - E_J^\star/\hbar} \, , \tag{E.1}$$

where $E_J^\star$ is the effective weak link Josephson energy. At short times then

$$G^0_{\varphi_0\varphi_0}(t) = 2i\alpha \ln\left(2\pi\alpha E_J^\star |t|/\hbar\right) + \dots \, , \tag{E.2}$$

where the omitted terms remain finite at small times. Using the bare propagator in the self-energy expression (17), the logarithmic divergence at small $t$ then gives

$$\Sigma(\omega) \simeq -iC \frac{E_J^\star}{2\pi\alpha} \left(\frac{\hbar|\omega|}{2\pi\alpha E_J^\star}\right)^{2\alpha-1} \, , \tag{E.3}$$

for $\hbar\omega \gg 2\pi\alpha E_J^\star$ and $\alpha < 1/2$, where $C$ is a constant with a positive real part. This power law would give the broadening of resonances a power-law frequency dependence $\sim \omega^{1-2\alpha}$ above the weak link resonance frequency. However, the above analysis ignores the finite ultraviolet cutoff of the physical system, typically set by the charging energy $E_C$ of the boundary junction. To see the predicted power law in an experimental realization would require a scale separation of several decades between $2\pi\alpha E_J^\star/\hbar$ and the ultraviolet cutoff, as well as performing linear response measurements at frequencies that are orders of magnitude smaller than the ultraviolet cutoff, that is typically in the $10^1$ GHz range, as illustrated by Fig. 10.

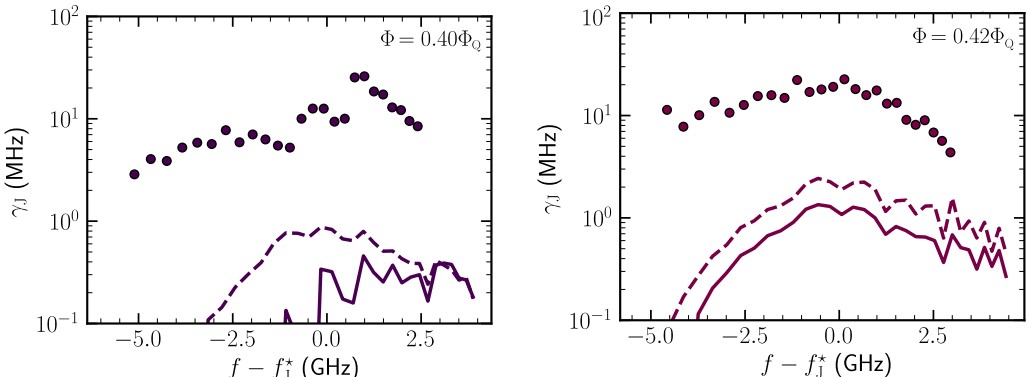

Figure 11: Nonlinear damping $\gamma_J$ as a function of the frequency for $\Phi/\Phi_Q = 0.40$ and 0.42. The dots correspond to the experimental data, the full lines are calculations from the perturbative treatment at second order, while the dashed ones correspond to the third order. At these lower magnetic fluxes, the circuit is in a regime where $\hbar\omega \sim E_J^\star$. Hence, the perturbative approach fails, as the discrepancy between the second and third order shows. However, the theory is still able to reproduce the maximum of damping when $\omega \sim \omega_J^\star$.

# F  Perturbative breakdown at intermediate $E_J^\star$

Data corresponding to the low frequency range $\hbar\omega < E_J^\star$ cannot be correctly described by the diagrammatic theory, as it is only valid for $\hbar\omega \gg E_J^\star$. While the calculations matches quantatively the experimental data for the large flux values where $E_J$ is small enough (see Fig. 6), we see in Fig. 11 that the theoretical predictions (full lines) underestimate the measured losses by an order of magnitude for two smaller flux values. In order to confirm the non-perturbative nature of this discrepancy, we pushed the perturbative expansion to third order in $E_J$.

In describing as simply as possible this higher class of diagrams, we only draw in what follows the diagrams with the lowest number of intermediate lines between each vertex, but have summed over all possible numbers of lines. This results in making the following replacements: $G_{\varphi_0\varphi_0}(t) \to \sin(G_{\varphi_0\varphi_0}(t))$ and $G_{\varphi_0\varphi_0}(t)^2/2 \to 1 - \cos G_{\varphi_0\varphi_0}(t)$. We also write here as double lines the full propagators of the skeleton expansion. The third order class of diagrams thus reads:

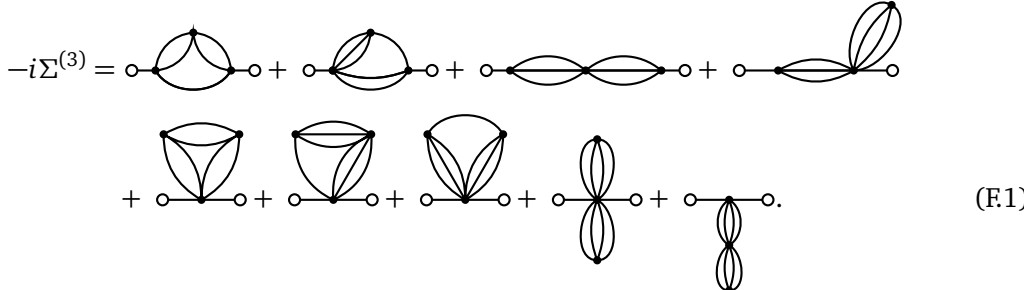

$$\tag{F.1}$$

Note that we did not include any nested diagram, since they are already generated by the skeleton expansion.

In Fig. 11 we compare the second order (full lines) and third order (dashed lines) diagrammatic results for the nonlinear damping rate at fluxes $\Phi = 0.40\Phi_Q$ and $\Phi = 0.42\Phi_Q$ with the corresponding experimental data (dots). There is a significant difference between the second and third order perturbative expansion implying that the expansion is not converged, although higher order corrections at least go in the right direction. At the larger fluxes of Fig. 6 where $E_J$

is markedly smaller, we find that the third order contribution is small compared to the second order one (provided $\hbar\omega \gg E_J^\star$), validating our theory.

The failure of the diagrammatic apporach comes from the fact that, when $\hbar\omega \sim E_J^\star$, the phase is partially trapped in the Josephson potential. Therefore, another source of damping, the phase slip between different minima of the Josephson potential, must be taken into account [44]. Since, the SCHA consists of replacing the cosine potential by an effective quadratic potential, these phase slip phenomena cannot be caught. However, although our theory is not quantitative in this regime, it correctly predicts that the maximum of $\gamma_J$ occurs at $\hbar\omega \sim \omega_J^\star$.

## G  Dielectric losses in the chain

In the remaining sections, we investigate whether more mundane loss mechanisms could provide an alternative explanation of our data. We start by considering the losses generated in the dielectric of the junction capacitances $C$ of the chain, that can be modeled by writing that $C = \left(\epsilon' + i\epsilon''\right)d$ where $\epsilon'$ and $\epsilon''$ are respectively the real and imaginary part of the dielectric permittivity while $d$ is a parameter proportional to the length which depends on the capacitance geometry. $\epsilon'$ gives the capacitive response of $C$ while $\epsilon''$ is its dissipative part. Hence, the admittance of $C$ is given by:

$$Y_C(\omega) = \frac{\omega}{i}\text{Re}(C) + \omega\text{Im}(C) = \frac{\omega C}{i} + \frac{1}{R_C(\omega)} \simeq \frac{\omega C}{i}\left(1 + i\tan\delta\right), \tag{G.1}$$

where $\tan\delta = \text{Im}(C)/\text{Re}(C) = \epsilon''/\epsilon'$. To find the damping induced by the dielectric, we use the dispersion relation (A.1), replacing $C$ by $C(1 + i\tan\delta)$ and taking the limit $ka \ll 1$ (which is equivalent to the mode number $n \ll N$, valid in the frequency windows that we probe). By defining the complex wavenumber $\kappa = ka$, and the dimensionless frequency $x = \omega l_c/v_\varphi$, with $l_c$ the screening length and $v_\varphi$ the velocity of plasma modes, we have:

$$\kappa^2 = \left(\frac{x}{l_c}\right)^2 \frac{1}{1 - x^2(1 + i\tan\delta)}. \tag{G.2}$$

Because of dielectric losses, the wavevector has a complex part: $\kappa = \kappa' + i\kappa''$. We suppose that losses are weak enough so that $\tan\delta \ll 1$ and hence $\kappa'/\kappa'' \gg 1$. At first order in these quantities, we find:

$$\kappa' = \frac{x}{l_c}\frac{1}{\sqrt{1 - x^2}}, \tag{G.3}$$

$$\kappa'' = \frac{\tan\delta}{2}\frac{x}{l_c}\frac{x^2}{(1 - x^2)^{3/2}}. \tag{G.4}$$

We then consider the small wavenumber limit $|\kappa| \ll 1/l_c$, that is equivalent to $n \ll N/l_c \sim 100$, which describes the modes below 10 GHz as seen in Fig. 7 (above this frequency the dispersion relation starts to bend). If $|\kappa| \ll 1/l_c$, then $x \ll 1$ and the chain behaves as an ideal transmission line sustaining TEM modes (see Eq. (G.3) with $x \ll 1$ ) and the quality factor of the modes are given by [59]:

$$Q_{\text{int}} = \frac{\kappa'}{2\kappa''}. \tag{G.5}$$

Since $2\pi Q_{\text{int}} = \omega/\gamma_{\text{int}}$ we end up with:

$$\gamma_{\text{diel}} = \frac{\omega}{2\pi}x^2\tan\delta. \tag{G.6}$$

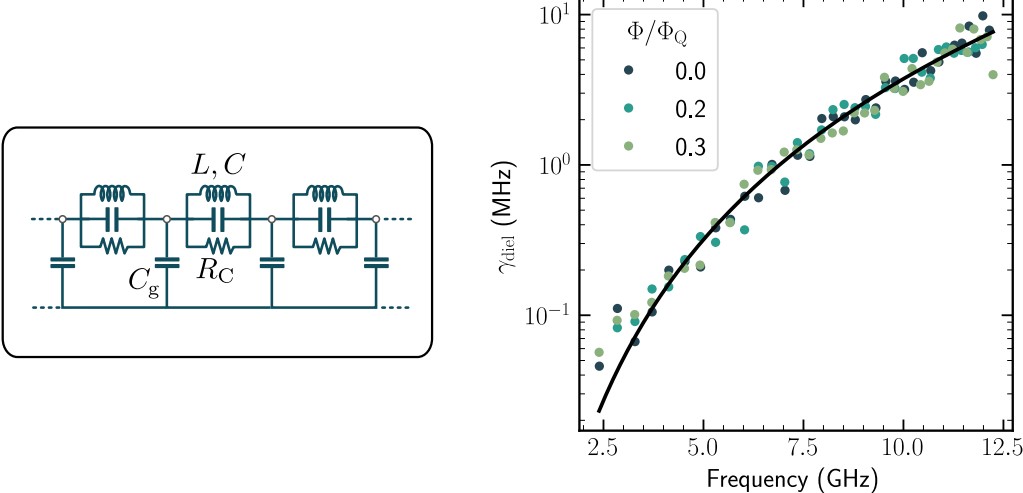

Figure 12: **Left.** Circuit used to model the dielectric losses in the chain. From this circuit we establish Eq. (G.2). **Right.** Dielectric damping $\gamma_{\text{diel}}$ as a function of the frequency. The dots correspond to the data measured for $\Phi/\Phi_{\text{Q}}$ equal to 0, 0.2 and 0.3 (blue to green). The black line is the result of the fit using Eq. (G.6).

Eq. (G.6) is used to fit simultaneously the internal damping for the magnetic fluxes $\Phi/\Phi_{\text{Q}}$ equal to 0, 0.2 and 0.3. For these three fluxes, the damping of the modes does not vary. Therefore, they do not appear to be caused by the SQUID nonlinearity. It has been noticed for chains of junctions [60] that $\tan\delta$ has a slight frequency dependence which can be parametrized as:

$$\tan\delta = A\omega^{b}\,, \tag{G.7}$$

where $A$ is the amplitude and $b$ should be close to unity. The results of the fit where $A$ and $b$ are the free parameters is given in Fig. 12. The good agreement between the model and the data shows that for these magnetic fluxes the damping of the modes are dominated by dielectric losses in $C$. From that fit, we estimate that $A \times 2\pi \times 1\text{GHz} = (3.4 \pm 1.5).10^{-4}$ and $b = (0.5 \pm 0.2)$. Hence, $\tan\delta \sim 10^{-4}$ in the gigahertz range, which is consistent with what is found in similar devices, confirming that the dielectric is a good suspect for the damping observed in this range.

## H   Losses at the boundary junction

In this section, we will investigate whether another mechanism could explain the observed damping of the chain modes, via the dissipation coming from the capacitive or inductive part of the SQUID at the boundary junction. We saw in the previous section that the dielectric used in the junctions of the chain can generate a damping. The same effect can take place at the level of the boundary junction, that we model by adding a resistance in parallel to the junction capacitance such that:

$$R_{\text{J,diel}}(\omega) = \frac{1}{\omega C_{\text{J}} \tan\delta}\,. \tag{H.1}$$

Since the SQUID itself is composed of two junctions, it can be expected that it can trigger damping of the circuit modes. On the other hand, since the circuit is superconducting, it is sensitive to quasiparticles. These quasiparticles can be modeled as a resistance is parallel to

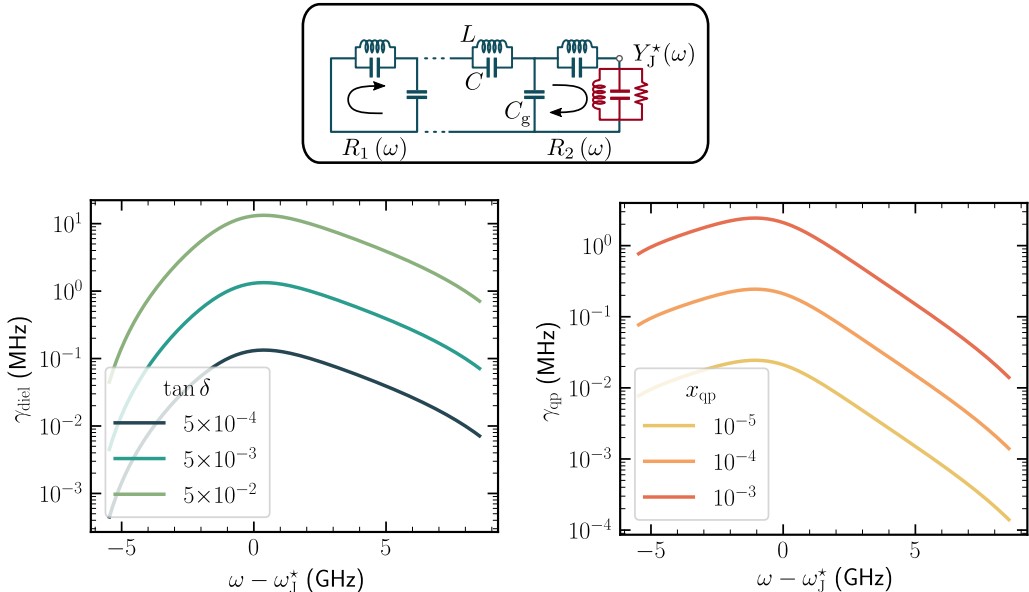

Figure 13: **Upper.** Circuit used to model the damping induced by an effective resistance at the SQUID site. $R_1$ and $R_2$ are the reflection coefficient in energy at site N and 0. $Y_J^\star$ is the admittance of the SQUID effectively described by a parallel RLC circuit. **Lower right.** Estimated damping $\gamma_{\text{diel}}$ as a function of $\omega - \omega_J^\star$ using Eq. (H.6), assuming dissipation comes from the SQUID dielectric. The damping are plotted for $\tan\delta$ ranging from $(5 \times 10^{-4})$ to $(5 \times 10^{-2})$ (blue to green). **Lower left.** Estimated damping $\gamma_{\text{qp}}$ as a function of $\omega - \omega_J^\star$, assuming dissipation comes from non-equilibrium quasiparticles. The damping are plotted for a quasiparticle density $x_{\text{qp}}$ ranging from $10^{-5}$ to $10^{-3}$ (yellow to orange).

the inductance of the SQUID. For quasiparticles in the high frequency regime, we have:

$$R_{\text{J,qp}}(\omega) = \frac{\pi\omega L_J^\star}{x_{\text{qp}}} \sqrt{\frac{2\Delta}{\hbar\omega}}, \tag{H.2}$$

where $x_{\text{qp}}$ and $\Delta$ are respectively the quasiparticles density normalized to the Cooper pair density and the superconducting gap of the superconducting material (taken as $\Delta = 210\,\mu\text{eV}$). In this modeling, we do not need to make hypothesis on the quasiparticles distribution. For both of these processes, the SQUID is modeled as a parallel RLC circuit where the inductance is $L_J^\star$, the capacitance $C_J$ and the resistance $R_{\text{J,diel}}$ or $R_{\text{J,qp}}$. We will now relate these losses at the boundary junction to the damping of the chain modes. To do this, we consider the circuit shown in the upper part of Fig. 13. The admittance of the effective RLC circuit at the boundary is:

$$Y_J^\star(\omega) = \frac{i}{\omega L_J^\star}\left[1 - \left(\frac{\omega}{\omega_J^\star}\right)^2\right] + \frac{1}{R_J(\omega)}. \tag{H.3}$$

Then, any wave propagating into the circuit will be reflected respectively by a coefficient $\sqrt{R_1}$ and $\sqrt{R_2}$ (in amplitude) at site N and 0, because of the impedance mismatch with the measurement line or with $Y_J^\star$. Let $E_{\text{em}}(0)$ be the electromagnetic energy stored in the circuit at a time $t = 0$. After a time $t_{\text{RT}} = 1/\Delta f_{\text{FSR}}$ (the round trip time) the energy in the circuit is given by:

$$E_{\text{em}}(t_{\text{RT}}) = |R_1(\omega)R_2(\omega)|E_{\text{em}}(0). \tag{H.4}$$

Hence, the energy decays exponentially with respect to time such that:

$$E_{\text{em}}(t_{\text{RT}}) = E_{\text{em}}(0)e^{-t/\tau_{\text{d}}}, \tag{H.5}$$

where $\tau_{\mathrm{d}}$ is the characteristic damping time in energy. This damping time is inversely proportional to the damping frequency, such that:

$$\gamma_{\mathrm{J}}(\omega) = \frac{\Delta f_{\mathrm{FSR}}(\omega)}{\pi} \ln \frac{1}{|R_1(\omega) R_2(\omega)|} . \tag{H.6}$$

Because we want to estimate the internal damping frequency, we consider that the reflection is perfect at site $N$, $|R_1(\omega)| = 1$, while the reflection at site 0 is given by:

$$R_2(\omega) = \left| \frac{1 - \tilde{Z}_{\mathrm{C}}(\omega) Y_{\mathrm{J}}^{\star}(\omega)}{1 + \tilde{Z}_{\mathrm{C}}(\omega) Y_{\mathrm{J}}^{\star}(\omega)} \right|^2 , \tag{H.7}$$

where $\tilde{Z}_{\mathrm{C}}$ is the characteristic impedance of the chain, where the plasma frequency is taken into account. Hence, $\tilde{Z}_{\mathrm{C}}(\omega) = Z_{\mathrm{C}}/\sqrt{1 - LC\omega^2}$. Therefore the internal damping at the junction site is given by

$$\gamma_{\mathrm{J}}(\omega) = \frac{\Delta f_{\mathrm{FSR}}(\omega)}{\pi} \ln \left| \frac{1 + \tilde{Z}_{\mathrm{C}}(\omega) Y_{\mathrm{J}}^{\star}(\omega)}{1 - \tilde{Z}_{\mathrm{C}}(\omega) Y_{\mathrm{J}}^{\star}(\omega)} \right| . \tag{H.8}$$

Now that we have an analytical formula relating $R_{\mathrm{J}}$ to the damping frequency, we can use it with Eq. (H.1) and (H.2) to estimate the corresponding loss $\gamma_{\mathrm{J}}$. The results are displayed in the middle panel (for dielectric losses) and lower panel (for the quasiparticles) of Fig. 13. The estimates are given for the circuit parameters corresponding to $\Phi/\Phi_{\mathrm{Q}} = 0.44$, one of the flux for which $\gamma_{\mathrm{J}}$ reaches its maximum.

We estimated the parameter $\tan\delta$ coming from the dielectric of the chain junctions to be about $10^{-4}$. There is no physical reason why that of the SQUID should differ from it by several orders of magnitude. However, we see in the middle panel of Fig. 13 that even taking an unrealistic value of $5.10^{-2}$, $\gamma_{\mathrm{J}}$ is underestimated the measured losses by about an order of magnitude. Finally, regarding the influence of non-equilibrium quasiparticles, the measured values ranges from $10^{-8}$ to $10^{-5}$. However, we see in the lower panel in Fig. 13 that even a quasiparticle density $x_{\mathrm{qp}} = 10^{-3}$ gives a $\gamma_{\mathrm{J}}$ that is off by two orders of magnitude. Hence, neither the dielectric nor the quasiparticles can realistically explain the order of magnitude of the measured mode damping $\gamma_{\mathrm{J}}$.

# I  Magnetic flux noise

The very convenient feature of a SQUID, its magnetic flux tunability has a price: its Josephson energy is sensitive to the noise of this control parameter. Since we saw that the frequency of the modes depends on the SQUID parameters, we expect the frequency of the modes to be time dependent. Hence, this noise could generate an inhomogeneous broadening of the modes. This broadening would then be more important where the modes are strongly influenced by the SQUID frequency, meaning those close to $\omega_{\mathrm{J}}^{\star}$. Furthermore, the broadening is also expected to be larger for larger noise in $E_{\mathrm{J}}$. Hence, using Eq. (6) we see that the broadening should be larger close to $\Phi/\Phi_{\mathrm{Q}} = 0.5$ where $E_{\mathrm{J}}$ depends strongly on magnetic flux (expect at the sweet spot induced by $d$ but since it is very small in our case the sweet spot is also extremely small). These two properties for such an inhomogeneous broadening are qualitatively compatible with our observations. On the other hand, this broadening would not give Lorentzian resonances (assuming a Gaussian noise). In any case, it will be interesting to check more quantitatively whether the flux noise in the terminal SQUID gives a negligible contribution to the broadening of the modes.

To estimate the flux broadening, we start by deriving the link between fluctuations in $E_{\mathrm{J}}$ and the fluctuation in the mode $n$ frequency, labeled respectively $\delta E_{\mathrm{J}}$ and $\delta\omega_k$. Where the fluctuation

of parameter $x$ is defined as $\delta x = \sqrt{\langle x^2 \rangle - \langle x \rangle^2}$. For the sake of simplicity we will consider that $\delta E_J \simeq \delta E_J^\star$. To estimate the error made using this approximation we can roughly estimate the $E_J$ over $E_J^\star$ dependency using the scaling formula Eq. (14). Using the estimated circuit parameter we find $E_J^\star \sim E_J^{\sqrt{2}}$. Consequently, $\delta E_J^\star$ should be slightly overestimated for small $E_J$ and slightly underestimated for large $E_J$. Hence for the flux close to $\Phi_Q/2$ we will slightly overestimate the frequency broadening and will therefore not be detrimental. Using Eq. (9) and (8), we have to first order:

$$\delta f_l(\Phi) = \frac{\Delta f_{\text{FSR},l}}{\pi} \delta(\theta_l(\Phi)), \tag{I.1}$$

where $\delta(\theta_l(\Phi))$ is the fluctuation of the phase shift and not the relative phase shift. Then, using the propagation of uncertainty and Eq. (B.10) we have:

$$\delta(\theta_l(\Phi)) = \left( \frac{(\hbar/2e)}{E_J^\star} \right)^2 \left| \frac{\partial \theta_l}{\partial E_J^\star}(\Phi) \right| \delta E_J^\star. \tag{I.2}$$

The partial derivative can be evaluated from Eq. (B.10):

$$\left| \frac{\partial \theta_l}{\partial E_J^\star}(\Phi) \right| = \frac{p(\omega)q(\omega)}{(1 - \omega^2 L_J^\star(\Phi)C_J)^2 + q(\omega)^2 \left( 1 - p(\omega)L_J^\star(\Phi) \right)^2}, \tag{I.3}$$

where:

$$p(\omega) = \frac{2}{L}\left( 1 - \omega^2 LC \right), \quad \text{and} \quad q(\omega) = \frac{\omega\sqrt{LC_g}}{2\sqrt{1 - \left( \frac{\omega}{\omega_p} \right)^2}}. \tag{I.4}$$

Now that we have an analytical formula relating $\delta f_l(\Phi)$ and $\delta E_J^\star$ we need to relate the latter to the magnetic flux noise. This can be simply done using Eq. (6). If we neglect the asymmetry of the SQUID here, we will overestimate the effect of the magnetic flux noise on $\delta E_J^\star$ close to half a quantum of flux but it makes the calculation easier:

$$\delta E_J^\star = \left| \sin\left( \pi\frac{\Phi}{\Phi_Q} \right) \right| \pi\frac{\delta\Phi}{\Phi_Q}. \tag{I.5}$$

Magnetic flux noise has been a topic of research since the pioneering works on DC SQUIDs [61]. Although much remains to be understood, it is quite well accepted that this magnetic flux noise (after appropriate filtering) seems to originate from spins at the junction interface [62], and can be modeled phenomenologically as a 1/f flicker noise:

$$S_\Phi(\omega) = A_\Phi^2 \left| \frac{2\pi}{\omega} \right|^\beta, \tag{I.6}$$

where $\beta \lesssim 1$ and $A_\Phi \sim 10^{-6} \times (h/(2e))$ [60, 63, 64]. Therefore, we can use Wiener-Khinchin theorem to estimate the magnetic flux noise:

$$\delta\Phi^2 = \frac{1}{\pi}\int_0^\infty d\omega S_\Phi(\omega). \tag{I.7}$$

This integral is obviously ultraviolet divergent. This is because such a noise is observed for a restricted frequency range, given by $f \in [10^{-4}\text{Hz}, 10^9\text{Hz}]$ [65]. However, since we are looking for a frequency broadening, the low frequency cutoff must by, at least, larger than the integration bandwidth used to acquire the $S_{21}$ data, otherwise we would be able to measure the

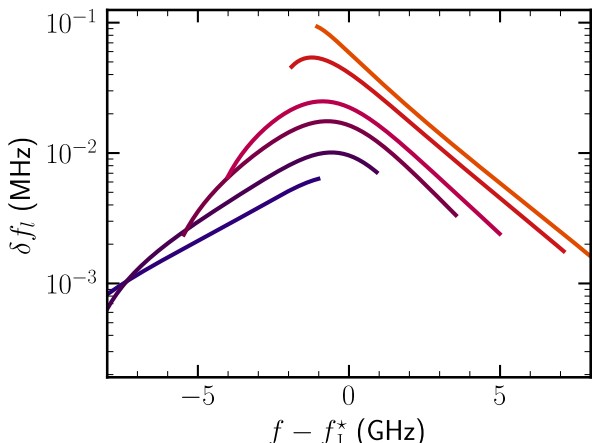

Figure 14: Estimated frequency homogeneous broadening $\delta f_l$ induced by a magnetic flux noise as a function of $f - f_J^\star$ for various magnetic fluxes between $0.35\,\Phi_Q$ and $0.5\,\Phi_Q$. The color code is the same than for Fig. 2 and 4.

time dependence of $\delta f_n(\Phi)$. We therefore set the low frequency cutoff to 1Hz. In addition, we set $\beta = 1$ for simplicity, as this will not influence future conclusions. With these assumptions we find:

$$\delta \Phi \sim A_\Phi \,. \tag{I.8}$$

Finally using Eq. (I.1), (I.3), (I.5) and (I.8) together, we can estimate the broadening induced by a magnetic flux noise. The result of this estimation is given in Fig. 14. This study shows that despite a qualitatively good behavior with respect to flux and frequency, the broadening is more than two orders of magnitude smaller than the measured $\gamma_J$. Hence, we can safely neglect the influence of a magnetic flux noise.

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
