# Peer review of "Revealing the finite-frequency response of a bosonic quantum impurity"

_SciPost Physics, doi:SciPost Phys. 14, 130 (2023)_

## Round 2 · Referee Report · Anonymous (Referee 1) · 2022-10-20

Strengths

  1. This work reports the results of a difficult experiment performed on a controllable quantum many-body system.

Weaknesses

  1. Lack of comparisons with the results of several earlier experiments performed with very similar systems.
  2. A part of the theoretical treatment is not reliable.
  3. Some of the qualitative considerations sound misleading or erroneous.
  4. The way several recent papers referred to is highly inadequate.

Report

This paper reports the measurements of the electromagnetic response of a small SQUID immersed in a high-impedance harmonic environment. The latter was constructed of relatively large-area Josephson junctions and may be thought of as a dissipationless transmission line supporting the propagation of microwave photons. The photons quantum fluctuations renormalize the SQUID's properties, bringing quantum many-body effects into the picture.

The main results of the work are represented by the experimental data of Figs. 5 and 6. The former figure shows the renormalized by quantum fluctuations Josephson energy EJ of the SQUID; the latter one details the dissipation of the transmission line photons caused by their decay into photons of smaller frequency.

The manuscript also contains some theoretical considerations focusing mostly on the Josephson energy renormalization in the case of "large" EJ and photon decay in the case of "small" EJ. The EJ-renormalization part contains little new (by authors' admission) but is an integral part of the data analysis. The other part is poorly formulated, contains some ad-hoc assumptions and tweaks performed just to get a better agreement with the experiment. Therefore, I doubt that section qualifies as a theory wielding any predictive power; in my view, it causes more harm than good (in part, because some of the qualitative considerations are misleading or erroneous).

The narrative of the manuscript is passable, except of a couple egregiously misrepresented references.

There are precious few experiments of the type performed by the Authors, so this work worth publishing in some form. However, the Authors must take care of several serious issues which I will address next.

Requested changes

  1. The results for the dissipation, Fig. 6, should be compared with the respective results of Ref. [45] (Fig. 3 therein).

  2. Quantify the definition of "large" EJ early in Section 4.1. For an unprepared reader, that becomes (sort of) clear only after seeing the in-line expression for the phase fluctuations preceding Eq. (14).

  3. Say something about EJ* in the beginning of Sec. 4.2. At the very least, tell it is not the one of Eq. (14).

  4. Calculation of the dissipation \gamma to the second order in EJ can be performed by standard means involving no more than the Gaussian averaging of exponentials linear in boson creation-annihilation operators (much like this is done in, e.g., theory of Mossbauer effect). Do that and present the result in a clear form in Sec. 4.2.
    Remove or relegate to a remote Appendix all of the currently present stuff involving the use of poorly defined (in my view) diagrammatic technique and tweaks such as "suitable regularization" with little physical meaning. The current content of p. 15 and a part of p. 16 does more harm than good. Here are two examples. The linearity of photon spectrum alluded to on p. 15 as important ingredient, is hardly important in a decay with no momentum conservation (as it is for the processes induced by a quantum impurity). The phase slips at low energy are suppressed (as the system flows towards the superconducting fixed point) and hardly relevant for the dissipation -- in an apparent contradiction to the statement on p. 16.

  5. The properly performed evaluation of \gamma should be sensitive to the time-reversal symmetry due to the photon transition selection rules that should be demonstrated to instill some confidence in the results.

  6. I am appalled by the way the work of Manucharyan's lab is represented in the manuscript. Clearly, the latter repeats the measurements reported in Ref. [45] on a circuit with somewhat different parameters. This is not spelled out in the manuscript under a lame excuse that Ref. [45] explores the "insulating" regime of BSG (see p. 1 of the manuscript). There are people among the Authors qualified enough to clearly understand: there is no qualitative difference between the "insulating" and "superconducting" regimes at \omega>> EJ*. On top of that, neither of the experiments is accurate enough to see the minute emerging differences between the regimes within the range of explored microwave frequencies. A later related work of the Manucharyan's group, https://doi.org/10.48550/arXiv.2203.17186, is simply not cited.

  7. I find the way Ref. [46] cited is misleading and does poor service to the community. Among the references to that paper scattered through the manuscript, the ones on p. 5 and p. 6 are the most damaging. The former one reads: "The presence of the secondary cutoff EC can affect quantitatively the general phase diagram of the BSG model [46]". This actually is an erroneous conclusion of Ref. [46], as a subset of the manuscript Authors explained in their recent comment https://arxiv.org/abs/2210.00742. (In addition to the quantitative arguments of that comment, one may recall that the phase diagram is entirely determined by the infrared physics oblivious to the specifics of an ultraviolet cutoff). The laudatory reference on p. 6, "Only recently [46] has a theoretical understanding started to emerge of the non-trivial physics ..." just helps to proliferate in the community an erroneous idea, which is challenged in the comment by the very manuscript's Authors. In my view, citations to Ref. [46] -- if kept -- should represent the views of the manuscript's Authors.

---

## Round 2 · Referee Report · Anonymous (Referee 2) · 2022-11-14

Report

This work deals with the experimental realization and investigation of an archetype bosonic quantum impurity system, the boundary sine-Gordon (BSG) model. Such system is realized in a superconducting platform where a small SQUID is galvanically coupled to a high impedance transmission line. The latter, being composed of 4250 Josephson junctions, simulates an Ohmic bosonic environment with high frequency cut-off wp. The impedance Z of the transmission line is designed to be a sizeable fraction of the quantum of resistance RQ. The SQUID is characterized by a flux-dependent Josephson energy EJ(phi); its internal capacitance CJ provides the second relevant energy scale, the charging energy EC. Finally, properties of the SBG system are evinced from measurements of the finite frequency response of the transmission line in various regimes of parameters.
Indeed major result of this work is the identification of different dynamical regimes of the impurity in a
EJ(phi)/EC vs w/wp plane. To this aim, the authors compare experimental data with analytical predictions for the dissipative response at high frequencies, and for the reactive response in the regime of moderate phase fluctuations. In the latter case a renormalization of the Josephson energy of the SQUID by the high frequency environmental modes is found to be in line with expectations for the ideal SBG model in a large parameter range. For deviations, the authors simply quote the very recent work by Masuka et al., Ref. 22, where a numerical renormalization group analysis is performed for a similar model. However, no comparison with predictions of that work is performed. This should be amended in a revised version of the paper.

In summary, this work represents a relevant and very timely endeavor in the field of quantum simulators. It confirms theoretical predictions on various parameter regimes of the BSG model and identifies open theoretical challenges. As such I highly recommend publication in SciPost. Before this, the authors should critically address the work by Masuka et al. in the context of their experimental results and theoretical analysis.

---

## Round 3 · Referee Report · Anonymous (Referee 4) · 2023-2-6

Report

The Authors addressed all of the points I raised in the previous report.

One of the sentences in their response ("The statement about no qualitative difference between the "insulating" and "superconducting" regimes is based on theories which become well-defined only in the limit omega -> 0 and astronomically large system sizes") indicates that they perhaps did not understand one of the points I made (unless "no" in the response is a typo). Nevertheless, in the resubmitted version they introduced adequate changes, which answer the respective part of the report.

Additional substantive changes in the text took care of all other recommendations made in the report.

In my view this manuscript may be published in its current form.

---

## Round 3 · Author Response

We thank the Referees for their comments and address them below.

Report of Referee 3

Referee 3 requests only that we compare further to the work of Masuki et al, (Ref. 46 in the first version of the manuscript; there is a typo in the report, which calls it Ref 22). Referee 2 also raises the fact that we cite Ref. 46. For the benefit of Referee 3, the issue is as follows: subsequent to us preparing the original version of the manuscript, some of us realised that the NRG analysis in Ref. 46 is wrong. See our comment: https://arxiv.org/abs/2210.00742. See therefore our response point 7 of referee 2's report.

Note further that a direct comparison is in principle not possible, since Ref. 46 is solely concerned with zero-frequency properties (the presence or absence of the Schmid transition), whereas we study finite frequency properties. See our response to point 6 of the report of Referee 2.

Report of Referee 2

We respond to the referee's recommendations point by point:

1) Fig. 3 of Ref [45] and Fig. 6 of our manuscript do not plot the same thing: Fig. 3 of Ref [45] shows gamma at input frequency equal to the resonance frequency omega_0, where it is maximal, while this resonance frequency is varied by varying the flux Phi. In the regime explored in Ref [45], a power law dependence (eq. 4 in that work) is expected for this plot. Our Fig. 6 shows gamma vs. input frequency at fixed flux, and hence FIXED omega_0. Both our theory and experiment yield an exponential dependence at fixed flux, which is what our figure is intended to illustrate. Our ability to compare to [45] is further limited by the fact that our device operates in a regime (alpha<1, and E_J<E_C) complementary to that of [45] (alpha>1 for theory and E_J>E_C).

Changes made:

In the new version, we have nonetheless compared our results to those in [45] as far as possible.

On p. 11 we now state:

"The individual data sets with Φ/Φ0 > 0.45 each show γJ decreasing by two decades as the probe frequency is scanned. This sharp frequency dependence within the physically accessible frequency window seems to be a unique property of the EJ ≪ EC regime. In Ref. [45], where EJ/EC > 1, γJ typically varies by less than one decade as the probe frequency is scanned." (This statement is based on Fig. 2 in [45].)

On p. 12 we now state: "In Ref. [45], similar round-trip decay probabilities are obtained in the transmon regime (1 < EJ/EC < 5) and α ≃ 2, while smaller decay probabilities were obtained in the transmon regime at α ≃ 0.7."

2) Indeed, in Sec. 4.1 we focus on values of E_J large enough so that the phase fluctuations (14) do not become much larger than unity.

Changes made: At the beginning of Sec. 4.1 on p. 12, we now say: "At given α < 1, fluctuations of the boundary phase φ0 are controlled by the ratio EJ/EC. Here we focus on the regime where EJ/EC is sufficiently large that phase fluctuations do not much exceed unity."

3) Indeed, the calculations of Sec. 4.2 concern the regime in which the emergent scale EJ* is not reliably estimated by Eq. (14).

Changes made: We have rearranged the text of Sec. 4.2 so that the part where the meaning of the emergent scale is explained appears at the start. We have also modified this explanation. The beginning of Sec. 4.2 on p. 14 reads: "For α < 1, the BSG model is known to flow to a Kondo-like strong coupling fixed point in the limit of zero temperature and for frequencies below a small emergent scale that characterizes the low-frequency inductive response of the weak link. The response of the system at these low frequencies are beyond the reach of a perturbative treatment [25]. Here we denote that scale EJ because in the universal regime ZJ ≫ Z, it has the same scaling as in Eq. 14. Note however that the device we are modelling is not in the universal regime. Nonetheless we may expect EJ < EJ."

4) The Referee's comment in fact concerns several points, so we address them one by one.

4a) Gaussian averaging of exponentials appearing in Eq.(36) can be performed in several equivalent ways, all giving the same result, and we are free to choose the one we like. We agree that the diagrammatic calculation is not the most compact way, but it has an advantage of showing the decay channels (see our response to point 5 below). Note that we also provide an independent non-diagrammatic calculation in addition to the diagramatic one. The perturbation theory is formally similar to that employed for the Cosine nonlinearity in the Bulk Sine Gordon model and the correctness of our expansion can therefore be checked against results in that context. See Figure 3 and Eq. 3.4 in Amit, Goldschmidt and Grinstein, J. Phys. A: Math. Gen. 13, 585 (1980), which we cite in the new version as Ref. [58].

Changes made:

In the revised version, we use the term "perturbation theory" instead "diagramatics" in Sec. 4.2 (which was probably the cause of the confusion). We also make it clearer in Appendix C that two independent derivations of Eq. 17 in the main text are provided, by putting each derivation in its own subsection. We have also reformulated parts of Sec. 4.2 and Appendix C to clear up confusion.

Below Eq. (18) on p. 15, we now state: "In Appendix C we present two independent derivations of this result."

In the introduction to Appendix C on p. 22, we now state: "We perform the same calculation twice, using two equivalent approaches. In both cases, we perform Gaussian averaging of exponents whose arguments are linear in bosonic creation and annihilation operators. In the first derivation, we perform the required normal ordering by hand using Wick’s theorem. In the second derivation, we represent the Wick contractions by Feynman diagrams. This is not as compact, but has the virtue of showing all multi-photon decay channels explicitly. The formal structure of our expansion is similar to that encountered for the bulk cosine nonlinearity in the Sine Gordon model so that the correctness of our result can be checked against results obtained in that context [58]."

We have also simplified the discussion in Appendix C by removing unnecessary detail in the form of two equations below Eq. 36, and replacing that section with "One can expand the sin and cos functions of the field operators into exponentials. Under time-ordering, the product of exponentials of field operators equals the exponential of the sum of the operators. One is thus left with evaluating <T exp X> where X is linear in boson creation and annihilation operators. It is well known that the result is <T exp X> = exp( <T X^2> /2)."

4b) Unfortunately, we cannot avoid discussing the regularization. We agree with the Referee that it has no physical meaning - it is just a technical trick - but it is necessary to define the whole procedure. When expanding in E_J around zero, as the Referee suggests in 4a, the Debye-Waller factor exp(- <phi_0^2> /2) is zero, due to a logarithmic divergence in <phi_0^2>. As a result, an unphysical answer gamma = 0 is obtained, so we do need to introduce a counter-term E_cutoff. Essentially, the Referee wants us to describe an ill-defined calculation procedure in the main text, and then refer the reader to an appendix to make this procedure meaningful. We think that our approach is more transparent.

Changes made: We have added the above sentences about the Debye-Waller factor and the counter-term to the text above Eq. 20 on p.15, but insist on keeping the discussion of the regularization as part of the main text.

4c) The near linearity of the photon dispersion is an important ingredient, even when there is no momentum conservation. The thing is that our chain has finite length, and linear dispersion results in equidistant spectrum of mode frequencies. The implication of this is explained further in a rewritten section at the bottom of p. 15. It explains why we had to use the self-consistent Born approximation which introduces many-body level repulsion and smoothens the self-energy. This is also a technical point, but we need it to justify the use of the self-consistent Born approximation (which formally goes beyond the second order in E_J).

Changes made: The rewritten text above Eq. 19 on p. 15 reads: "Owing to the approximately linear dispersion relation of the array ω = vk in the ohmic regime, single-photon resonance frequencies are almost equally spaced. As a result there is a large near-degeneracy in these multi-photon resonances. For instance, if we denote the lowest bare resonance frequency by ω1, then there are 16 multi-photon resonances, each involving an odd number of photons, at frequency 10ω1, which corresponds to a single photon resonance in the middle of the experimentally accessible frequency window. This leads to a highly singular behaviour of the self-energy in the vicinity of these degenerate clusters of multi-photon resonances, when it is built on bare Green’s functions. The situation significantly worsens at finite temperature, where resonances involving absorption of thermal photons must be taken into account. In our device this is not mitigated appreciably by the slight curvature of the photon dispersion or by geometric irregularity [46]. This singular behaviour is however spurious as it does not take into account the significant many-body level repulsion between multi-photon states coupled directly or indirectly by the highly non-linear terminal junction."

4d) In the superconducting phase, the phase slip contribution indeed dies out at frequencies well below the scale EJ, but before doing that it ruins the perturbative calculation at frequencies ~ EJ (if one does not use an appropriate regularization). The cause of trouble was probably the poorly formulated sentence in the end of page 3, which we removed in the revised version. But we see nothing wrong in what we wrote about phase slips on pages 15,16. Still we did try to reformulate that discussion to make it more clear.

Changes made: We removed the sentence "In the low frequency regime below the renormalized scale, we find that diagrammatic theory breaks down, due to enhanced phase slips contributions" that used to appear on p.3. The discussion that the referee refers to on pages 15,16 of the previous version now appears on p.17 and reads in the new version: "At smaller fluxes, the superconducting phase is trapped near minima of the periodic Josephson potential, and non-perturbative 2π-phase slip processes between minima provide the dominant contribution to the damping process, which are not taken into account in our perturbative treatment. Deviations from our model thus gives an estimate of these phase slip processes at α<=1. These have also been investigated theoretically and experimentally at α>=1 [43–45]."

5) If we understand correctly, the Referee's remark about the time-reversal symmetry refers to the fact that the Hamiltonian is even in the phases, and in particular in phi_0. In the new version, we explain how the resultant photon number parity conservation manifests in our Eq. 17.

Changes made: Below Eq. 44 in Appendix C we included the following explanation: "Because the Hamiltonian is even in the phases, and in particular in φ0, photon number parity is conserved, that is, one photon can decay into an odd number of photons only. This selection rule is fully respected by our calculation, and the diagrammatic representation of the self-energy is especially convenient to see this fact. Namely, non-linear vertices can have only an even number of legs, and in the sin(G) − G self-energy term which is responsible for the decay, each vertex (0 or t) has one and only one external leg, so the two vertices must be connected by an odd number of photonic lines."

6) Contrary to the Referee's statement, our manuscript does not repeat the measurements of Ref. [45]. Firstly, our current work is the culmination of our own earlier works [39,40]. In [40] we noticed the following: "A more surprising observation is that the odd modes are much more damped than the even ones. We interpret this as resulting from the non-linearity that odd modes inherit from the small Josephson junction. This is experimental evidence of the strong back-action of the small Josephson junction on the many modes of the chain forming its linear environment." Our present work was motivated by the need for a device in which the broadening seen in [40] could be resolved optimally. Secondly, we explore a different parameter regime to [45]. This brings about different physics: while in Ref. [45] the photon decay is mainly controlled via phase slips, our device is dominated by the perturbative physics. Finally, in Ref. [45] the boundary junction is in the transmon regime with a well-defined resonance of its own, while in our device the boundary junction resonance is completely dissolved in the continuum.

The statement about no qualitative difference between the "insulating" and "superconducting" regimes is based on theories which become well-defined only in the limit omega -> 0 and astronomically large system sizes, and whose applicability to the real device is not obvious; so experiments performed (presumably) in a different phase are still valuable. As for arXiv.2203.17186, we indeed forgot to cite it, and we thank the Referee for pointing this out.

Changes made: We now mostly avoid mention of "superconducting" or "insulating" regimes, preferring instead simply "alpha<=1" or "alpha>=1" when we discuss the finite frequency response of the system.

We contextualise our work in the last sentences of the second last paragraph of p. 3 as follows: "Our work complements several recent experimental and theoretical studies [41–45] that target non-linear effects in a bosonic impurity model at impedances equal to or larger than the superconducting resistance quantum RQ = h/(2e)^2 ≃ 6.5 kΩ. Of particular relevance is [45] in which the relatively large Josephson to charging energy ration EJ/EC ratio of the terminal SQUID combined with an environmental impedance larger than RQ put the device in a regime where the losses are dominated by phase slips. We target here the exploration of quantum non-linear effects at some- what lower dimensionless impedances α = Z/RQ ≃ 0.3 and at EJ/EC < 1, where quartic (Kerr-type) and higher order processes at the terminal junction are the dominant source of non-linear effects. In addition, the reactive response of the SQUID is studied in parallel. This allows to measure and explain both sides of the same problem for the first time with this type of system."

We further discuss [45] in the first paragraph on p.4 as follows: "Ref. [45] studies a complementary regime where the EJ/EC ratio of the terminal junction is larger than 1. At α~>1, Ref. [45] matches measured many-body losses to theoretical predictions. Viewed together with our work, this demarcates the regime of α~<1 and Josephson energy comparable to charging energy as the frontier for further theoretical work or quantum simulation."

We cite arXiv.2203.17186 as Ref. [46] in the new version. It is mentioned in the last sentences on p.3: "A bosonic impurity can couple single and multi-photon states of a multi-mode resonator. The resulting appearance of multi-photon resonances in linear response has been demonstrated experimentally [46]."

7) We prepared this manuscript before it became clear to some of us that the former Ref. [46] was wrong, and we forgot to update on this point in the latest version. The Referee is absolutely right. We do not want to propagate incorrect findings and have removed mention of the former Ref. [46] from our manuscript.

---

## Round 3 · List of Changes

Please see authors comments above.

---

## Editorial Decision

published